# Fine optimization of a dissolution dynamic nuclear polarization experimental setting for $^{13}$C NMR of metabolic samples

**Arnab Dey[1], Benoît Charrier[1], Karine Lemaitre[1], Victor Ribay[1], Dmitry Eshchenko[2], Marc Schnell[2], Roberto Melzi[3], Quentin Stern[4], Samuel F. Cousin[5], James G. Kempf[6], Sami Jannin[4], Jean-Nicolas Dumez[1], and Patrick Giraudeau[1]**

[1]Nantes Université, CNRS, CEISAM UMR 6230, 44000 Nantes, France
[2]Bruker Biospin, Industriestrasse 26, 8117 Fällanden, Switzerland
[3]Bruker Biospin, Viale V. Lancetti 43, 20158 Milan, Italy
[4]Université de Lyon, CNRS, Université Claude Bernard Lyon 1, ENS de Lyon, Centre de RMN à Très Hauts Champs (CRMN), UMR5082, 69100 Villeurbanne, France
[5]Aix Marseille Univ., CNRS, ICR, 13397, Marseille, France
[6]Bruker Biospin, 15 Fortune Dr., Billerica, MA 01821 USA

**Correspondence:** Patrick Giraudeau (patrick.giraudeau@univ-nantes.fr)

**Abstract.** NMR-based analysis of metabolite mixtures provides crucial information on biological systems but mostly relies on 1D $^1$H experiments for maximizing sensitivity. However, strong peak overlap of $^1$H spectra often is a limitation for the analysis of inherently complex biological mixtures. Dissolution dynamic nuclear polarization (d-DNP) improves NMR sensitivity by several orders of magnitude, which enables $^{13}$C NMR-based analysis of metabolites at natural abundance. We have recently demonstrated the successful introduction of d-DNP into a full untargeted metabolomics workflow applied to the study of plant metabolism. Here we describe the systematic optimization of d-DNP experimental settings for experiments at natural $^{13}$C abundance and show how the resolution, sensitivity, and ultimately the number of detectable signals improve as a result. We have systematically optimized the parameters involved (in a semi-automated prototype d-DNP system, from sample preparation to signal detection, aiming at providing an optimization guide for potential users of such a system, who may not be experts in instrumental development). The optimization procedure makes it possible to detect previously inaccessible protonated $^{13}$C signals of metabolites at natural abundance with at least 4 times improved line shape and a high repeatability compared to a previously reported d-DNP-enhanced untargeted metabolomic study. This extends the application scope of hyperpolarized $^{13}$C NMR at natural abundance and paves the way to a more general use of DNP-hyperpolarized NMR in metabolomics studies.

## 1 Introduction

NMR spectroscopy offers unparalleled robustness and reproducibility for the analysis of complex metabolite mixtures. Such advantages make NMR an ideal tool for a number of analytical applications such as targeted or untargeted metabolomics, stable-isotope-resolved studies of metabolism, pharmacokinetic studies, and bioprocess optimization (Zhang et al., 2010; Wang et al., 2013; Zhang et al., 2008; Calvani et al., 2010; Liu et al., 2010; Strickland et al., 2017; Emwas et al., 2019; Kim et al., 2013; Wishart, 2008). However, NMR suffers from poor sensitivity, which limits the detection of metabolites to the micromolar concentration range (in contrast, the limit of detection of mass spectroscopy can reach sub-nanomolar concentrations; Grotti et al., 2009; Liem-Nguyen et al., 2015; Li et al., 2020). Ow-

ing to such a challenge, the analysis of metabolic mixtures by NMR mostly relies on $^1H$ spectroscopy, which is, however, often marred by the strong signal overlap in $^1H$ spectra of complex biological samples due to limited spectral dispersion. $^{13}C$ NMR could be a promising solution as it offers wide spectral dispersion, which results in a better separation of metabolite signals. At present, the application of $^{13}C$ NMR to metabolite mixtures at natural abundance is limited due to about 2900-fold reduced sensitivity (owing to its low natural abundance and gyromagnetic ratio) versus $^1H$. Therefore, to expand the applicability of $^{13}C$ NMR metabolomics, it is of much interest to develop methods which improve the sensitivity of $^{13}C$ signal detection while retaining its resolution advantage. Indeed, detecting major metabolites in biological samples at natural abundance would require reaching signal-to-noise ratio (SNR) values above 10 for millimolar (mM) concentrations or less, which is not possible with conventional hardware. The development of homemade $^{13}C$ optimized NMR probes for metabolomics has been shown to yield improved accuracy in the metabolite identification and group separation for mass limited samples (Clendinen et al., 2014). However, the $^{13}C$ signal sensitivity generally remains too low for routine metabolomics applications, and further development is required to improve sensitivity.

Hyperpolarization techniques are in the forefront among such developing methods. Hyperpolarization stems from creating a far from equilibrium spin population distribution, which results in a significant increase of the nuclear spin polarization compared to thermal equilibrium values, leading to considerable improvement in sensitivity. Several hyperpolarization techniques such as dissolution dynamic nuclear polarization (d-DNP; Jannin et al., 2019; Giraudeau et al., 2009; Singh et al., 2021; Dey et al., 2020; Guduff et al., 2017; Leon Swisher et al., 2015; Dumez et al., 2015), parahydrogen-induced polarization (PHIP; Kiryutin et al., 2019; Ivanov et al., 2009), and its reversible version, signal amplification by reversible exchange (SABRE; Lloyd et al., 2012; Daniele et al., 2015; Eshuis et al., 2014; Guduff et al., 2019), CE1 have been implemented successfully for the analysis of complex mixtures. Among all the hyperpolarization techniques, d-DNP is of particular interest for metabolic mixtures as it has been known to improve the signal sensitivity by more than 10 000 times in a nonselective fashion (Ardenkjær-Larsen et al., 2003).

In d-DNP, nuclear spins are polarized in the solid state at cryogenic temperatures (typically 1–2 K), in a high magnetic field (3–7 T), by microwave irradiation in the presence of a radical species. This is followed by rapid dissolution and transfer of the sample to a nearby NMR spectrometer, where hyperpolarized signals are acquired at room temperature in the liquid state. Despite offering impressive sensitivity improvements, the instrumental complexity of d-DNP could appear contradictory with the high throughput, precision, and robustness needed for analytical applications. However, several recent studies highlighted the potential of d-DNP for an-

alyzing complex metabolic mixtures by $^{13}C$ NMR. Lerche et al. (2018) TS1 demonstrated the relevance of d-DNP for fluxomic studies by quantifying the $^{13}C$ isotopic patterns to understand the metabolic activity of cancer cell extract incubated with $^{13}C$-enriched glucose (Frahm et al., 2021; Lerche et al., 2018; Frahm et al., 2020). Recent investigations also focused on the development of d-DNP methods to analyze metabolic samples at natural $^{13}C$ abundance, taking advantage of $^1H \rightarrow ^{13}C$ cross-polarization (CP) in the solid state to reach high $^{13}C$ polarization levels in a short time (Batel et al., 2014; Bornet et al., 2013). In 2015, we showed that d-DNP could be used to detect metabolites of plant and cell extracts at $^{13}C$ natural abundance (Dumez et al., 2015), and we then reported that the repeatability of this approach was compatible with metabolomics applications (Bornet et al., 2016a).

In a recent proof-of-concept study, we demonstrated that d-DNP could be incorporated into a full untargeted metabolomics workflow capable of separating tomato extract samples at two different ripening stages and of highlighting corresponding biomarkers (Dey et al., 2020). In this study, we described preliminary experimental optimizations that played a key role in achieving the precision needed for the application of d-DNP to a series of metabolic samples. These included the use of a Hellmanex™-coated NMR tube which helps to reduce the formation of microbubbles due to the rapid motion of the dissolved sample inside the NMR tube. Moreover, we showed that with the use of an appropriate internal standard, the effect of instrumental variability on relative signal quantification could be reduced from about 10 % to 3 %. Overall, the study showcases the potential of d-DNP for metabolomics. However, this study also highlighted that a full utilization of the prototype DNP setting for such application would require a thorough optimization of several experimental parameters. Such optimization was beyond the scope of that report owing to the large number, complexity and interdependence of parameters involved in the d-DNP setting. A recent review by Elliot et al. (Elliott et al., 2021) provides a detailed description of the practical aspects of the d-DNP workflow, highlighting the good experimental practices that ensure optimized sensitivity and line shape on the resulting liquid-state spectra. In the perspective of applying d-DNP to complex diluted mixtures of metabolites at $^{13}C$ natural abundance, a particular focus should be made on the experimental parameters that impact sensitivity and repeatability, two key ingredients for analytical applications. Such optimization should also be oriented towards potential users of this equipment, who may not be experts in instrumental development.

In this context, the present study demonstrates a systematic optimization of a prototype d-DNP setting, focusing on the parameters which could be tuned by the user without the need of instrumental development. More specifically, we present the results of a fine optimization of d-DNP settings for the analysis of metabolic mixtures at $^{13}C$ natural abundance, showcasing a significant improvement (about 5 times

in the quaternary $^{13}$C region and 50 times in protonated $^{13}$C region) in sensitivity compared to our previous study while preserving a high repeatability. We also intend this optimization procedure to serve as a guideline for the various applications of d-DNP in the field of analytical chemistry.

## 2 Design of experiment

A brief schematic description of the d-DNP experimental setting is presented in Fig. 1, highlighting the most important components. The figure represents the operational workflow indicating important steps in sequence. The complete d-DNP operation is divided into four main experimental steps, i.e., sample preparation, polarization in the solid state, dissolution and transfer, and signal acquisition in the liquid state. In the operational workflow, we have schematically indicated the change of several relevant parameters as experienced by the sample during the d-DNP experiment.

Table 1 provides a list of all parameters involved in the workflow, indicating those to be optimized from a practical user viewpoint without the expertise of hardware development. In the next section, we qualitatively describe all the relevant parameters involved in DNP experimental setting and identify the parameters which could potentially impact the analytical performance of the d-DNP workflow for the analysis of metabolic mixtures at natural $^{13}$C abundance.

Most of these parameters are interdependent. Therefore, instead of a sequential, one-by-one optimization of these parameters, the ideal strategy would consist in testing all (or many) possible parameter combinations. However, it would require a number of experimental attempts that would be unrealistic with respect to the liquid helium consumption, considering that three experiments for each condition would be required to evaluate the repeatability. Therefore, we have divided all the parameters into four main operational subunits that we optimized in two layers, i.e. (i) a systematic investigation with no repetition to find out the optimum combinations of impactful parameters on a sensitivity basis and (ii) an evaluation of the repeatability on the basis of three experiments for the most sensitive conditions.

## 3 Experiments and parameters

In this section, we sequentially describe the different steps of the experiment, highlighting the key parameters in the perspective of application to the sensitive and repeatable analysis of metabolite mixtures at natural $^{13}$C abundance. CE2

### 3.1 Sample preparation (a)

The d-DNP sample preparation procedure is essential to enable uniform nuclear spin polarization across the sample at a cryogenic temperature, which in turn affects the achievable sensitivity and repeatability (El Daraï and Jannin, 2021). A standard sample preparation protocol is required with a careful choice of parameters. Based on previous d-DNP studies and on our own experience, we identified three potentially important parameters that could impact the d-DNP workflow and should be optimized: "DNP juice" composition, ripening time, and polarizing agent (PA) concentration (Plainchont et al., 2018; Köckenberger, 2014; Elliott et al., 2021). For the experimental optimization and evaluation of the d-DNP workflow, a mixture of three common metabolites at natural $^{13}$C abundance (L-alanine, sodium acetate, and sodium pyruvate) was prepared, each at a 5 mM concentration, which is representative of the concentration of major metabolites in extracts. Sodium 3-trimethylsilylpropionate-d4 (Na-TSP-d4; 98 % D; 20 mM) was added as an internal standard, as previously reported (Dey et al., 2020). For DNP experiments, these chemicals were dissolved in a glassy matrix along with the polarizing agent (PA), 4-hydroxy-2,2,6,6-tetramethylpiperidine-1-oxyl (TEMPOL). To ensure optimal solubility, samples were stirred for 60 s with a mechanical stirrer. Sample sonication was also evaluated by sonicating the sample for 60 s before inserting the sample into the polarizer, but it did not impact the polarization efficiency.

For the whole study, samples were prepared from the same stock solution to avoid unwanted variation from the differences in sample measurement. The stock solution was prepared by solubilizing the metabolites and TEMPOL in the DNP juice, then DNP samples were equally divided and stored inside a $-80$ °C freezer. Before the start of the polarization experiment, samples were taken out from the freezer and stirred at room temperature as mentioned above and then transferred into the sample cup. Care was taken to avoid small bubbles and residue of sample droplets residing in the top part of the sample cup, which is above the active $\mu w$ irradiation region. The sample cup was then vitrified at ca. 4 K inside the polarizer.

It is also important to trace the amount of sample (in weight) taken inside the sample cup before hyperpolarization. In our case, 200 µL of DNP sample weighed 258 mg, with a standard variation of 1 % of all the samples used for experiments. Also, we have monitored the $^1$H signal integral without microwave at the solid state to investigate the variation of signal integral without microwave. It is important to note that such a signal integral differs from the actual thermal signal of the sample as the acquired signal integral includes the background signal from the sample cup.

In the following section, we discuss the key parameters involved in sample preparation for d-DNP experiments.

### 3.1.1 PA concentration (a.1)

The PA plays a central part in DNP polarization. A broad variety of PAs is available depending on the targeted sample and application. It has been well discussed in several studies that nitroxide-based radicals such as TEMPOL are preferred for $^1$H $\rightarrow$ $^{13}$C CP-based d-DNP, as the broad EPR

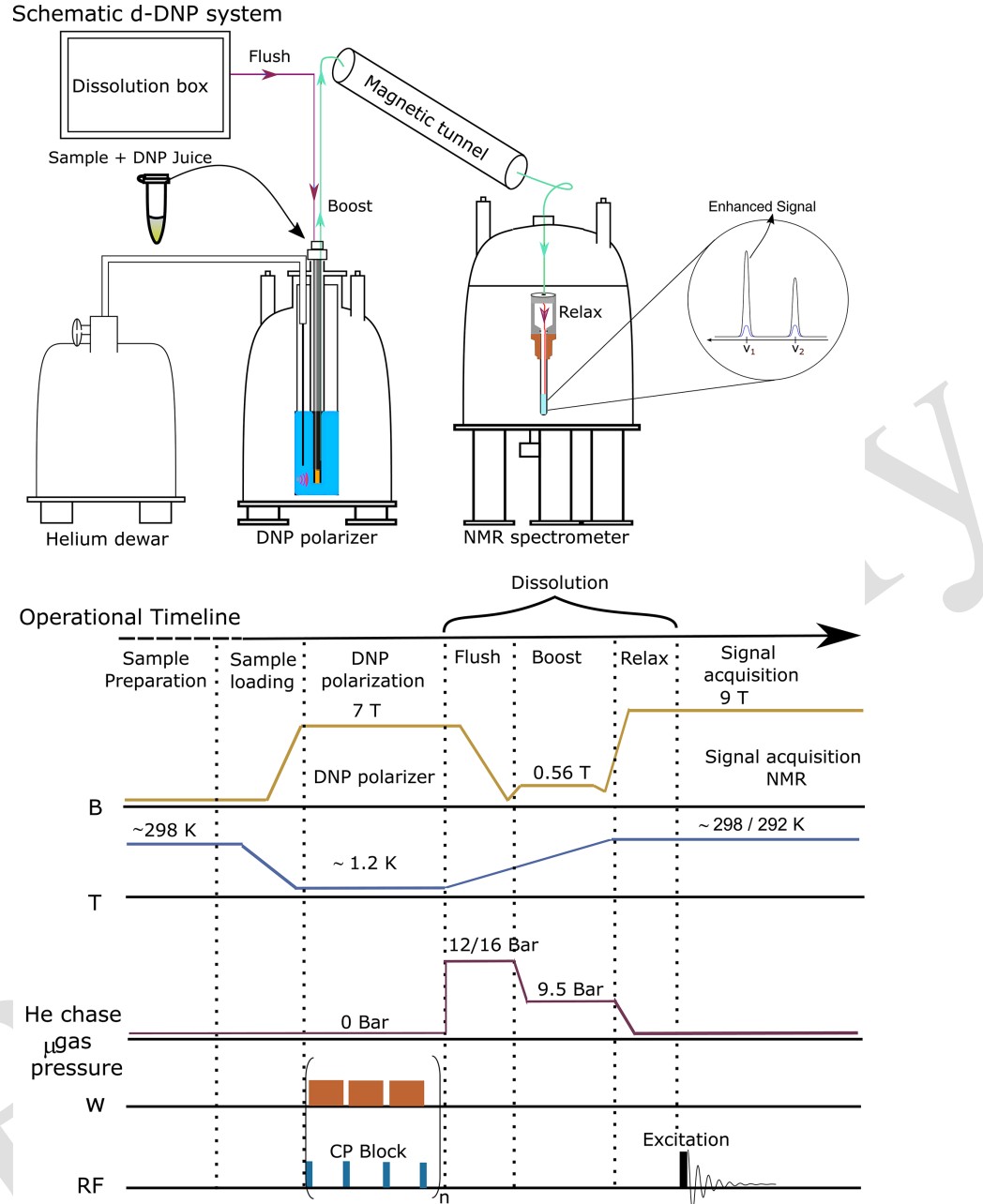

**Figure 1.** Schematic description of the semi-automated prototype dissolution DNP system along with operational timeline, indicating important events during d-DNP operation in sequence. TS2

linewidth of TEMPOL ensures high and rapid $^1$H polarization, which in turn is the main source of $^{13}$C polarization. Therefore, TEMPOL was chosen as a PA for this study. A range from 25–100 mM concentration of TEMPOL has been previously investigated (Elliott et al., 2021). For the application to metabolite mixtures at natural $^{13}$C abundance, the TEMPOL concentration should be optimized to achieve the highest DNP polarization through a rapid DNP buildup with minimal contribution to the polarization losses during sample transfer before liquid-state signal acquisition.

### 3.1.2 DNP juice composition and order of mixing (a.2)

The DNP juice composition consists of a mixture of glycerol, $D_2O$, and $H_2O$, which ensures a uniform distribution of analytes and polarizing agent (PA) forming a glassy sample at cryogenic temperatures (about 1–2 K). Previous studies reported that the efficiency of DNP juice particularly de-

**Table 1.** List of parameters for the plan of optimization of DNP setting. PA: polarizing agent. CP: cross-polarization. $\mu w$: microwaves. VTI: variable temperature insert (enclosure where the sample is vitrified, hyperpolarized, and dissolved at cryogenic temperature).

| | (a) Sample preparation | (b) DNP polarization | (c) Dissolution | (d) Signal acquisition |
|---|---|---|---|---|
| Parameters optimized | (1) PA concentration (2) DNP juice composition and order of mixing (3) Ripening time ($R_T$) (4) Sample sonication parameters | (1) $\mu w$ parameters (2) $^1$HDNP buildup (3) CP parameters (4) Vitrification parameters | (1) Dissolution solvent (CD$_3$OD, D$_2$O) (2) Dissolution duration (3) Sample transfer line | (1) Pre-shimming & Pre-tuning |
| Parameters not optimized | – Choice of PA – Thermal history of the sample | – VTI temperature & pressure stability – Field strength of the DNP polarizer – $\mu w$ source | – Dissolution solvent volume – Dissolution temperature & pressure – Other dissolution solvent – Other sample transfer methods | – Standard $^{13}$C NMR parameters (pulse length, power, decoupling, etc.) |

pends on the nature and concentration of PA (Leavesley et al., 2018). Several studies reported that a glycerol content $> 55\%$ was sufficient to form a glass at 1–2 K (Puzenko et al., 2005; Leavesley et al., 2018; Hayashi et al., 2005). However, a very high percentage of glycerol in the DNP juice restricts the solubility of the biological sample in the DNP juice. Previous DNP studies reported up to 60 % of glycerol content in the DNP juice composition (Jähnig et al., 2019; Tran et al., 2020; Overall and Barnes, 2021; Kaushik et al., 2022). Care should be taken to decide the order of mixing the sample in H$_2$O–D$_2$O and glycerol of DNP juice depending on the solubility in H$_2$O–D$_2$O compared to glycerol for ensuring complete solubility of the sample in DNP juice.

### 3.1.3 Ripening time (a.3)

A recent study reported that following the completion of DNP sample preparation, a delay before vitrification (ripening time) could lead to the formation of nanoscopic water vesicles in a glycerol rich matrix, resulting in an inhomogeneous distribution of PA in the two water and glycerol phases of the DNP juice (Weber et al., 2018). Such nanoscopic phase separation was reported at a PA concentration of 10–80 mM, which could hamper the $^1$H DNP efficiency by 20 %. The optimum ripening time ($R_T$) was reported to depend on the sample, polarizing agent, and DNP juice composition. Therefore, it is essential to investigate the impact of ripening time for diluted metabolite samples at natural $^{13}$C abundance.

### 3.2 DNP polarization (b)

In this section, we describe the relevant instrumental details of the polarizer including the cryostat along with the microwave source, followed by a discussion on the parameters involved in the optimization that impact repeatability and sensitivity.

*DNP polarizer.* CE3 The prototype Bruker d-DNP polarizer works at field (7.05 T) and temperature (1.15 K), which offers optimal CP-based capabilities to reach high $^{13}$C polarization levels in about 15 min (Bornet et al., 2013; Dey et al., 2020). It is built on a standard 7.05 T wide-bore magnet and cryostat design, modified to accommodate a variable temperature insert (VTI). The VTI enables DNP at 1.15 K, using liquid helium (*l*-He) introduced from a transport dewar (e.g., 100 L) and custom transfer line into a phase separator (PS) near the top of the VTI. From there, a membrane pump (Vacuubrand MD 4 NT) transfers cold gaseous helium (*g*-He), whose enthalpy cools the neck, baffles, and radiation shields of the VTI, while *l*-He flows down from the PS and enters the sample space via automated needle valves near the VTI tail. A main pump (Edwards iGx600L) acts on the admitted *l*-He for final cooling of the sample space, whose temperature setpoint is chosen via a feedback-controlled butterfly valve to the pump. For DNP, the microwave source consists of a synthesizer (8–20 GHz) and an amplifier and frequency multiplier chain (AMC; Virginia Diodes, Inc) to deliver a final frequency of $\sim 198$ GHz at $\sim 120$ mW. A waveguide carries the $\mu w$ into the VTI to irradiate the sample. Frequency modulation (Bornet et al., 2014) is programmed via the low-frequency source, while microwave gating (Bornet et al., 2016b) is achieved via TTL (transistor–transistor logic) pulses from the Bruker AV NEO NMR console to the AMC. For NMR, the two-channel console runs Topspin 4 and is coupled to a custom Bruker $^1$H,$^{13}$C probe, with an external tuning and matching (room temperature) for an overall circuit able to achieve simultaneous nutation frequencies of 50 kHz without arcing.

*Optimization.* The efficiency of the DNP polarization depends on the instrumental design ($\mu w$ source, cryostat, polarizer, radio frequency (RF) coil, etc.). Therefore, the ability of users to improve sensitivity and repeatability is limited. Our system was designed to offer a highly repeatable polarization

in the solid state, and further instrumental modifications are beyond the scope of this study. However, the sensitivity and repeatability are also impacted by user-dependent parameters, such as the polarization temperature and the $\mu w$ and CP parameters, which are further described below.

### 3.2.1 Microwave parameters (b.1)

It is essential to find out the optimal $\mu w$ irradiation frequency and power as well as associated modulation bandwidth to achieve optimal polarization. The optimal value of such parameters depends on the temperature and sample formulation. Here, $\mu w$ optimization was performed at 1.2 K and for the optimal sample preparation parameters.

### 3.2.2 [1]H DNP buildup (b.2)

The measurement of [1]H polarization buildup rate helps to verify the PA's integrity and also dictates the optimum $\mu w$ irradiation time (which is chosen to be once or twice the [1]H DNP buildup time) between "contact" for polarization transfer from [1]H to [13]C via CP. For each sample, before polarizing the [13]C spins, the [1]H polarization buildup time was measured using the pulse sequence shown in Fig. B1a at 1.2 K.

### 3.2.3 Vitrification parameters (b.4)

Formation of a glass during vitrification inside the polarizer is important to obtain repeatable polarization. Care should be taken while inserting the sample to maintain a similar rate of vitrification in the cryostat. It is important to note that in some cases we experienced a sudden drop of [1]H polarization buildup time in spite of an identical sample composition, which resulted in a reduction of [1]H and [13]C DNP signal integrals. This could be due to the impact of the sample insertion rate on the formation of glassy matrix at cryo-temperature inside the cryostat. However, such reduction did not impact the liquid-state signal integral, and we concluded that the vitrification rate did not impact our results. Still, to limit potential associated effects, we took care of keeping the same sample insertion time (40 s) inside the cryostat for all experiments. Also, dissolving the metabolites and PA in $H_2O$ and $D_2O$ followed by dissolving the resulting solution in glycerol helped to improve the solid-state signal repeatability compared to the reverse sequence of dissolving the metabolites and PA in DNP juice (first in glycerol then in $H_2O$ and $D_2O$).

### 3.2.4 CP parameters (b.3)

As discussed in the Introduction, achieving [13]C hyperpolarization via cross-polarization (CP) from DNP-polarized [1]H spins is the key for metabolomics application. The pulse sequence implemented to polarize [13]C nuclei is presented in Fig. B1b. The optimization of CP parameters and the methodological developments ensuring efficient CP have been described thoroughly in previous studies (Elliott et al.,

2021). Here, we followed a similar procedure of optimization and implemented these developments for our study.

### 3.3 Dissolution (c)

After the completion of [13]C hyperpolarization at 1.2 K, the hyperpolarized sample is rapidly dissolved in a hot, pressurized solvent, followed by a rapid transfer to the liquid-state spectrometer through a magnetic tunnel to minimize polarization losses due to the nuclear spin relaxation at room temperature during transfer. There are a number of developments aiming for a rapid and robust dissolution process, such as the development of gas-driven and liquid-driven sample transfer systems (Katsikis et al., 2015; Ceillier et al., 2021; Bowen and Hilty, 2010), a built-in sample transfer system attached to a cold sample cup (Kress et al., 2021), and solid sample transfer (Kouřil et al., 2019). Each of the methods have their own advantages and disadvantages which have been reviewed in detail (Elliott et al., 2021). Here, we focus on the optimization of the gas-driven dissolution system available on our setup.

In our case, dissolution is achieved upon manual coupling of a fluid transfer stick to the sample cup after it has been lifted ($\sim$ 9 cm) just above the *l*-He level. The stick includes two parallel capillaries (PEEK; 1.6 mm ID): an inlet for the preheated, pressurized dissolution solvent and an outlet to carry hyperpolarized fluid via a sample transfer line to a 5 mm NMR tube situated in the probe of the solution-state NMR observation magnet. The hyperpolarized solid sample is dissolved in 5 mL of hot solvent, and the helium gas drives the dissolved liquid inside the transfer line to run through a 0.56 T magnetic tunnel (Milani et al., 2015) (DNP Instrumentation, https://dnp-instrumentation.com, last access: 25 September 2022). Inside the liquid-state NMR spectrometer, a passive receiver system (injector) accepts the turbulent dissolution sample and then facilitates phase separation (liquid sample and helium gas) and settling through gravity after introduction of the sample into the NMR tube at ambient temperature and pressure.

In this section, we discuss the experimental parameters related to the optimization of the dissolution, transfer, and relax steps. In our previous study, the long duration of this process (time from the start of the dissolution to the start of the signal acquisition = 11.3 s) significantly reduced the sensitivity of [13]C metabolite signals (Dey et al., 2020). Moreover, the dissolution step contributes most to the signal unrepeatability as it involves a manual step. Therefore, careful optimization of the dissolution is crucial to ensure the maximum and repeatable amount of hyperpolarization before signal acquisition. From a technical point of view, the dissolution process consists of three main events: (i) flushing the pre-pressurized hot solvent to the sample cup for certain duration (termed "flush", driven by the pressure difference between the pressure cooker and the sample space), (ii) pushing (using helium gas) the dissolved hyperpolarized sample for a fixed period

of time (termed "boost") through the sample transfer line to reach to the injector, and (iii) collecting the liquid and allowing the pushing helium gas to be released (termed "relax" duration) before the dissolved sample reaches the connected NMR tube. The relax time ensures the liquid is filled at least up to the active RF coil length devoid of any microbubble, and to limit the residual motion of liquid that would impact the line shapes. A longer delay has a favorable impact in the improvement of signal line shape and linewidth, however resulting in sensitivity losses due to the polarization decay which impacts differently depending on the relaxation of different $^{13}$C sites. The optimum value of the delay needs to be decided upon balancing the two opposing effects mentioned above to obtain better $^{13}$C signal sensitivity for the majority of metabolites. Also, this delay depends on the physical properties (viscosity, surface tension etc.) of each dissolution solvent. Note that the relax delays contain a fixed duration delay (0.1 s trigger), which is required to switch/trigger the automatic signal acquisition pulse sequence in a liquid-state spectrometer.

The scheme indicating three different stages of dissolution process is shown with different colors in Fig. 1. There are several parameters involved in these three events which can be optimized to reduce the loss of polarization. We focused on the optimization of the following parameters:

1. dissolution solvent (choice of solvent, solvent volume, dissolution pressure, dissolution temperature) (c.1) TS3

2. dissolution duration (duration of flushing, boosting, and relaxing) (c.2)

3. sample transfer line (length and inner diameter) (c.3).

It is important to note that all parameters listed above are correlated to each other. Therefore, we have focused on optimizing the combination of parameters instead of optimizing parameters one by one. Before presenting our attempts to find the best combination of parameters, we introduce the influence of these parameters in the context of maximizing the available hyperpolarization in the liquid state.

### 3.3.1 Dissolution solvent (c.1)

The dissolution solvent has a significant impact on the efficiency of dissolving the hyperpolarized solid at a 1.2 K temperature and on the speed of sample transfer as well as on stabilizing the dissolved liquid inside the NMR tube. $D_2O$ is widely accepted as a dissolution solvent due to its high heat transfer coefficient, which leads to efficient dissolution of the hyperpolarized solid sample. Also, higher solubility of metabolites or other biological samples in $D_2O$ forms an extra advantage. However, owing to its higher viscosity and surface tension, $D_2O$ is less efficient in terms of sample transfer speed, and a longer stabilization delay is required to avoid microbubble during signal acquisition. Methanol-$d_4$ has been known to be used as an alternative dissolution

solvent to boost the sample transfer rate and to reduce the stabilization delay as it is less viscous and has lower surface tension compared to $D_2O$ (Singh et al., 2021; Mishkovsky and Frydman, 2008). Both dissolution solvents have their own advantages and disadvantages. For example, the potential solubility of metabolites depends on the heat transfer efficiency, which in turn depends on the specific heat capacity and on the solvent temperature set at the dissolution box. Such specific heat capacity could be smaller for methanol-$d_4$ compared to $D_2O$ (the specific heat capacity of water and methanol is about 4.18 TS4 and 2.53 kJ$(kg\,K)^{-1}$ respectively; for $CD_3OD$ and $D_2O$ we set the temperature in the dissolution oven at 156 and 170 °C respectively). We decided to determine the best combination of dissolution parameters for both solvents. Note that the choice of solvent is limited to these two options, owing to the incompatibility of the sample transfer material and the bad solubility of metabolites in other solvents.

The dissolution solvent volume influences the overall signal sensitivity in the liquid state. Reducing the solvent volume decreases the dilution factor, which may either increase or decrease of signal sensitivity depending on the relative influence of two opposing effects originating from the higher radical concentration in the dissolved sample vs. increase of sample spin concentration. However, a sufficient amount of dissolution solvent is necessary to efficiently dissolve the hyperpolarized solid inside the polarizer at a temperature of $\sim 1$ K. In our system, the dissolution solvent volume (5 mL) was already optimized by the instrument provider.

Dissolution temperature and pressure also play a role in efficiently dissolving and transferring the dissolved liquid. The choice of temperature is limited by the boiling point of the solvent at a particular pressure. Also, the choice of pressure is limited by the integrity of dissolution components. Therefore, these two parameters will be fixed as initial settings considered a "safe" maximum value for our dissolution setup.

### 3.3.2 Dissolution duration (c.2)

The optimum combination of flush, boost, and relax durations is essential to reduce the overall dissolution time. The flush duration mainly impacts the sample melting process, the boost duration is responsible for transferring the dissolved sample through the sample transfer line, and the relax duration is required to release the propellant helium gas avoiding microbubbles in the liquid-state sample before signal acquisition. Among all three durations, the boost and relax duration have the highest impact on fast sample transfer and improved signal line shape respectively. Therefore, optimization of the boost time and relax time is of highest priority in the optimization.

### 3.3.3   Sample transfer line (c.3)

The inner diameter (ID) and length of the sample transfer line influence the speed of sample transfer from the polarizer to the signal acquisition spectrometer and, also, influence the formation of bubbles in the dissolved sample. In our present setup, two different IDs (1.575 and 2.375 mm) of sample transfer line were available. We investigated the effect of ID of sample transfer line on the liquid-state signal sensitivity.

## 3.4   Signal acquisition (d)

Upon completion of the dissolution process, the liquid sample is collected in the NMR tube by gravity, and the pulse sequence automatically triggers to start signal acquisition, after a relax delay (discussed above). All d-DNP-enhanced NMR experiments were recorded at room temperature on a 400 MHz Bruker AVANCE NEO spectrometer equipped with a liquid-N$_2$ cryogenically cooled probe (5 mm CryoProbe™ Prodigy BBFO with ATMA and Z-gradient from Bruker BioSpin) using standard optimized pulse sequence and calibrated pulse parameters. The $^{13}$C spectra were recorded in a single scan at a 90° flip angle using Waltz-16 $^1$H decoupling during acquisition. They were processed with 1 Hz Lorentzian line-broadening, zero-filled to 256 000 data points, Fourier-transformed, manually phase-corrected, and automatically baseline-corrected with a polynomial of degree 5.

### Pre-shimming and pre-tuning (d.1)

Due to the rapidly decaying and irreversible nature of hyperpolarization, the method does not allow tuning and shimming to be performed before acquisition of the signal on the hyperpolarized liquid sample. Therefore, the hyperpolarized signal is acquired on pre-tuned and pre-shimmed condition. Pre-tuning and pre-shimming are done using similar sample composition and maintaining a similar sample height in the NMR tube as in the case of hyperpolarized signal acquisition. Moreover, to achieved improved line shape, it is desirable to perform pre-shimming at the similar temperature as the temperature of hyperpolarized, dissolved liquid during signal acquisition. Note that to optimize the quality of experiments in methanol, we have acquired a $^1$H spectrum of the residual protonated methanol using the same dissolution settings with methanol-d$_4$ and calculated the temperature of hyperpolarized liquid after injection (292 K) from the $^1$H chemical shift difference of the methyl and –OH groups. Then, the pre-shimming of identical sample was done at 292 K, which helps to improve the line shape of $^{13}$C signals. We noticed that DNP signal acquisition at 292 K improves linewidth of $^{13}$C signals of metabolites by more than 11 % compared to the DNP signal acquired at 298 K.

## 4   Results and discussion

In this section, we describe the result of parameter optimization under each subunit in the experimental sequence of events (sample preparation, polarization, dissolution, acquisition). In this section, each parameter is mentioned using the numbering defined in Table 1. Also, while comparing liquid-state signals at different parameter optimization stages, we analyzed only those signals which are above the limit of quantification (SNR > 10). Figure 2 shows d-DNP-enhanced $^{13}$C spectra of the metabolite mixture along with the reference, before experimental optimization of parameters as used in the previous report (Dey et al., 2020). Note that apart for TSP which is more concentrated, only quaternary carbons are visible due to their low signal-to-noise ratio (SNR).

### 4.1   Sample preparation (a)

### 4.1.1   PA concentration (a.1)

We compared three potentially suitable concentrations of TEMPOL (i.e., 75, 50, and 25 mM). The $^1$H DNP polarization buildup time for 75, 50, and 25 mM of TEMPOL was 20, 53, and > 3600 s respectively (Fig. C1). Here, to maintain high-throughput conditions, we compared the DNP polarization of $^{13}$C at different TEMPOL concentration with a fixed $^{13}$C DNP polarization time of about 20 min (using contact time of 15 ms, which was found to be optimal for each radical concentration and 80 s of $\mu w$ irradiation per each cycle of polarization transfer $^1$H → $^{13}$C (CP contact)). Figure 3 compares DNP-enhanced $^1$H and $^{13}$C signal in the solid state, as well as liquid-state $^{13}$C signal integrals at the same polarization duration. Note that the differences of signal integral at different radical concentration do not quantitatively reflect the polarization due to bleaching effect in solid state (Stern et al., 2021). Nevertheless, at a 50 mM radical concentration, the solid-state as well as liquid-state signals are particularly more sensitive than other TEMPOL concentrations at a fixed experimental time. It is important to note that although the protonated carbon of TSP shows slightly higher sensitivity at 25 mM TEMPOL due to smaller relaxation losses during dissolution, the sensitivity of other $^{13}$C signals is considerably lower compared to 50 mM TEMPOL. At 25 mM TEMPOL, it would be possible to achieve similar DNP polarization as at 50 mM TEMPOL but at the cost of an experimental time an order of magnitude higher.

### 4.1.2   DNP juice composition (a.2)

We have investigated the DNP efficiency of two different compositions of DNP juice (5 : 4 : 1 and 6 : 3 : 1 glycerol-d$_8$ : D$_2$O : H$_2$O, $v/v$), which have been reported to be efficient conditions for polarization with nitroxide based radicals. As noted in the previous section, further increase in

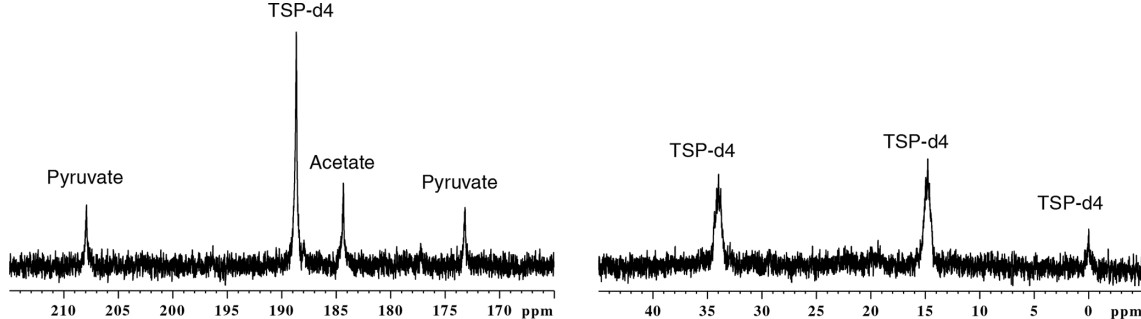

**Figure 2.** The d-DNP-enhanced $^{13}$C–{$^1$H} spectra of metabolites acquired before the optimization, indicating all the relevant signals above the limit of detection.

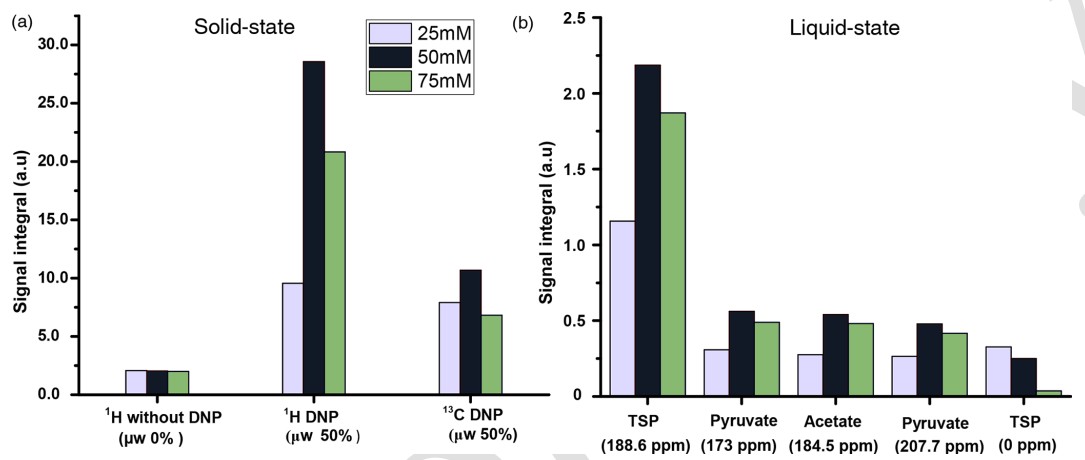

**Figure 3.** Comparison of signal integral values of **(a)** $^1$H and $^{13}$C signals in the solid state and **(b)** $^{13}$C–{$^1$H} liquid-state signal integrals of metabolites with a 25, 50, and 75 mM TEMPOL concentration.

glycerol content would result in an insolubility of metabolites in the DNP juice. It is worthwhile to note that in our previous DNP-based metabolomic work, the composition of the DNP juice was $5 : 4 : 1$. Figure 4a compares the $^1$H and $^{13}$C signal integral values at two different DNP juice compositions, which shows that DNP juice composition of $6 : 3 : 1$ (glycerol-$d_8$ : $D_2O$ : $H_2O$, $v/v$) offers higher polarization compared to the $5 : 4 : 1$ (glycerol-$d_8$ : $D_2O$ : $H_2O$, $v/v$) with similar repeatability. Further increase in glycerol content could reduce the solubility of metabolites which may hinder the metabolomic application in general. Therefore, the optimized DNP juice composition will be $6 : 3 : 1$ for the rest of the study.

### 4.1.3 Ripening time (a.3)

With our sample of choice and DNP juice, we did not find any considerable change of polarization after 30 min of ripening time (defined as the sum of time elapsed from sample preparation to insertion in the freezer and from the freezer to insertion inside the polarizer), as reflected in Fig. 4b. Therefore, we chose to systematically wait 30 min at room temperature before vitrifying the sample inside the DNP po-

larizer and also to prepare all the samples at the same room temperature to avoid unnecessary sources of signal variation.

### 4.2 Polarization (b)

#### 4.2.1 Microwave optimization (b.1)

In Fig. A1, we show the evolution of the relative $^1$H signal integral vs. $\mu w$ frequency and power. From this plot, we have chosen 198.08 GHz as an optimized frequency, which corresponds to the negative DNP polarization and 50 % of available $\mu w$ power for CP-based $^{13}$C polarization. We also investigated the effect of the $\mu w$ modulation bandwidth for efficient $\mu w$ excitation by observing the $^1$H signal integral at different modulation bandwidths and frequencies. The optimum values (a triangular frequency modulation with a bandwidth ($\Delta f_{\mu w}$) of $\pm 5$ MHz and frequency of 10 kHz is used) remained unchanged from our previously reported studies.

#### 4.2.2 $^1$H DNP buildup (b.2)

We measured the $^1$H buildup time of our sample at a 50 mM TEMPOL concentration with optimum DNP juice composi-

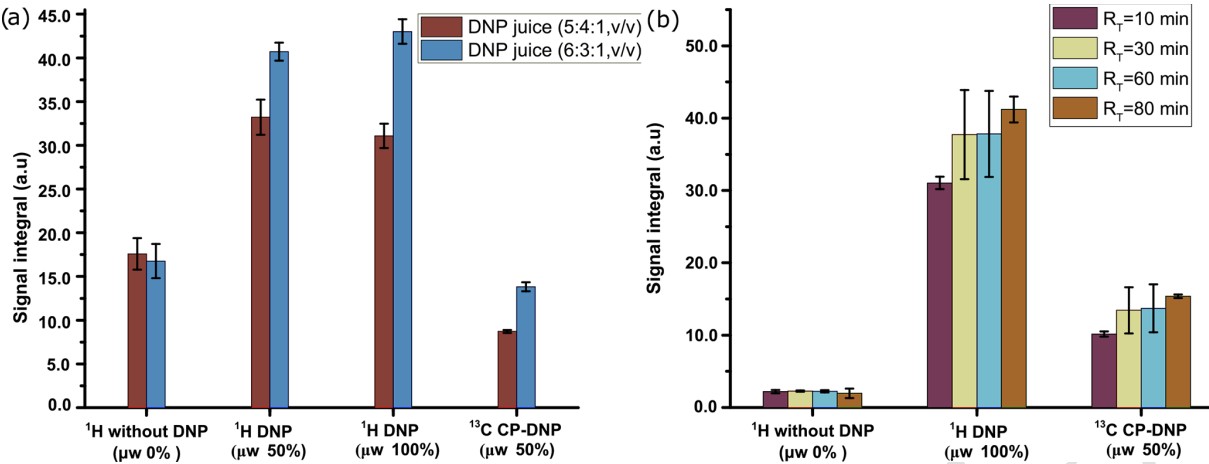

**Figure 4. (a)** Plot of solid-state $^1$H signal without $\mu w$ as well as $^1$H and $^{13}$C DNP signal integrals at two different DNP juice compositions with 50 mM TEMPOL using optimized $\mu w$ parameters. **(b)** Plot of solid-state $^1$H and $^{13}$C signal without $\mu w$ as well as DNP signal integral at different ripening time with same TEMPOL concentration using optimized $\mu w$ parameters. The standard deviation for every average integral value is calculated from three identical samples and displayed as an error bar.

tion at 1.2 K, leading to an estimated value of 53 s with 5 % variation over successive experiments.

### 4.2.3   CP parameters (b.3)

After optimization, the $^1$H polarization is transferred to $^{13}$C by 16 CP contacts of 15 ms each at intervals of 80 s, with a RF power of 15 W on $^1$H (using rectangular pulse with constant RF amplitudes of 21 kHz) and 60 W on $^{13}$C (using ramped up pulse with linearly increasing RF amplitudes from 16 to 23.2 kHz). Adiabatic half-passage pulses (WURST) of 30 and 60 W (pulse duration of 175 µs, sweep width of 100 kHz) were used on $^1$H and $^{13}$C channels respectively before and after the CP contacts. The total duration of CP experiment was 21 min.

### 4.3   Dissolution (c)

#### 4.3.1   Dissolution timing optimization (c.2)

CE4Before optimization, the duration of flush, boost, and relax times were set to 0.2, 5, and 6.1 s. We have considered this total duration of dissolution time (flush, boost, and relax, 11.3 s) as an upper limit with the objective to reduce the duration in the optimization process. Among the three durations, the boost time is the most critical duration for optimization. Therefore, we first focused on comparing several boost durations by analyzing the $^{13}$C signal of metabolites with a fixed set of flush time (0.2 s) and relax time (2.1 s). The results presented in Fig. 5 were obtained using a capillary transfer line with 1.575 mm inner diameter and a length of 370 cm with D$_2$O as a dissolution solvent.

The comparison shows improved signal integral values as the boost time is reduced from 5 to 1 s. Due to the reduction of boost time, the fast-relaxing protonated $^{13}$C signal

(the protonated $^{13}$C signal of TSP at 0 ppm) shows significant improvement compared to the quaternary $^{13}$C, but protonated $^{13}$C signals from other metabolites remain invisible. However, SNR comparison in Fig. 5b indicates an optimum boost time of 2 s corresponding to an improved line shape. A similar comparison with methanol-d$_4$ solvent exhibits the same boost time duration of 2 s for optimal sample transfer.

We have compared the repeatability of the newly optimized boost time with the repeatability before optimization (Fig. 6). Figure 6 shows improved signal integrals (especially for the protonated $^{13}$C of TSP) while retaining similar repeatability at the optimized dissolution duration. Following the optimization of the boost time, we also tested different flush times (data not shown) at a fixed boost duration and relax duration of 2 s and 1.1 s respectively. Overall, it was found that the reduction of flush durations did not improve signal sensitivity for both dissolution solvents.

As mentioned earlier in the experimental section, the relax time optimization is crucial to have an improved spectral line shape of hyperpolarized $^{13}$C signal with minimum loss of polarization. In the following section, we show the relax time optimization result at a fixed flush duration and boost duration of 0.2 and 2 s, respectively, for the two dissolution solvents (methanol-d$_4$, D$_2$O) separately as this optimization is solvent-specific.

#### 4.3.2   Dissolution solvent (c.1)

With D$_2$O as dissolution solvent, the protonated and the quaternary $^{13}$C signal of TSP at 2.1 s of relax time offers significant improvement of sensitivity compared to other relax values (see Fig. 7a). Here, considering a significant sensitivity improvement of $^{13}$C signals, we set 2.1 s as an optimum relax time. With CD$_3$OD, we obtained optimum sensi-

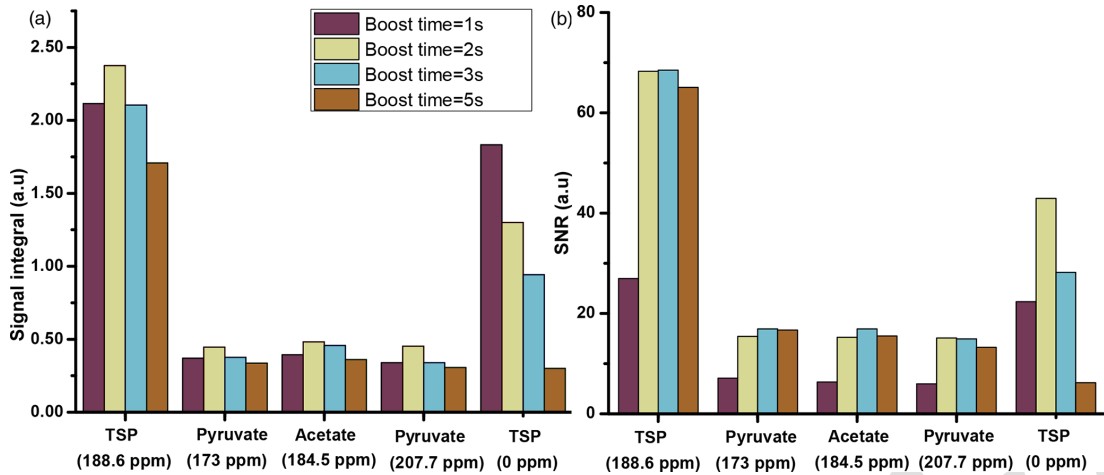

**Figure 5.** Plot of liquid-state hyperpolarized $^{13}$C–{$^{1}$H} signal integrals **(a)** and SNR **(b)** of the metabolites at different boost times using optimized $\mu w$ and solid-state DNP parameters using a transfer line with 1.575 mm ID and D$_2$O as dissolution solvent. The optimum value of boost time is chosen to be 2 s.

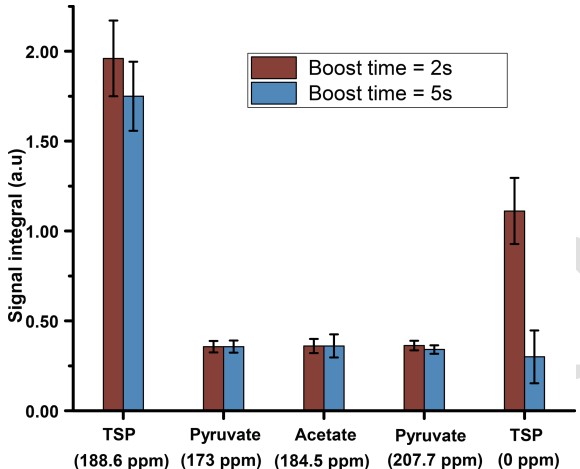

**Figure 6.** Plot of liquid-state hyperpolarized $^{13}$C–{$^{1}$H} signal integral repeatability before and after optimized dissolution timings using 1.575 mm ID of transfer line and D$_2$O as dissolution solvent, keeping a flush and relax delay of 2.1 s.

**Table 2.** $T_1$ values of metabolites in D$_2$O in the presence and absence of TEMPOL.

| Metabolites | Relaxation time ($T_1$, s) | |
| --- | --- | --- |
| | With TEMPOL | Without TEMPOL |
| Pyruvate (207.7 ppm) | 13.6 | 22.8 |
| TSP-d$_4$ (188.6 ppm) | 6.1 | 35.2 |
| Acetate (184.4 ppm) | 12.4 | 51.8 |
| Alanine (177.2 ppm) | 12.3 | 28.2 |
| Pyruvate (172.6 ppm) | 14.3 | 41.0 |
| Acetate (26 ppm) | 6.2 | 10.4 |
| Pyruvate (28.9 ppm) | 6.4 | 11.7 |
| TSP-d$_4$ (0 ppm) | 3.0 | 5.2 |

tivity at 1.1 s of relax time despite the irregular behavior of the quaternary $^{13}$C of alanine at 4.1 s and protonated $^{13}$C of TSP (see Fig. 7b). Indeed, with CD$_3$OD, the dissolved sample stabilizes more quickly compared to D$_2$O, owing to the lower viscosity and surface tension of CD$_3$OD. Although at relax time = 1.1 s we obtained optimum SNR with CD$_3$OD, we experienced some random failures in signal acquisition. Systematic investigation of this failure revealed that caution needs to be taken at the connection point of the injector and NMR tube to avoid failure in acquiring the signal at 1.1 s of relax time.

As indicated in Fig. E1, the NMR tube should be exactly connected to the bottom end of the injector as the imperfect connection at the junction between the injector and NMR tube causes inefficient filling of liquid in the NMR tube before the start of signal acquisition. This often results in failure of signal acquisition. We have designed a special gauge to ensure proper positioning of the NMR tube in the injector, which completely solved such a failure issue. In a nutshell, the optimum total duration of the dissolution time (time from the start of the dissolution to the start of the signal acquisition) was set to 4.3 and 3.3 s considering flush, boost, and relax durations of 0.2, 2, and 2.1 s for D$_2$O and 0.2, 2, and 1.1 s for CD$_3$OD respectively. To appreciate the impact of dissolution time optimization, we have measured the $T_1$ values of metabolites in the presence (at final concentration of TEMPOL after dissolution) and absence of TEMPOL, as presented in Table 2.

These $T_1$ measurements were done by dissolving the metabolites (each at 500 mM concentration) in D$_2$O, with similar composition of DNP juice present in the post-

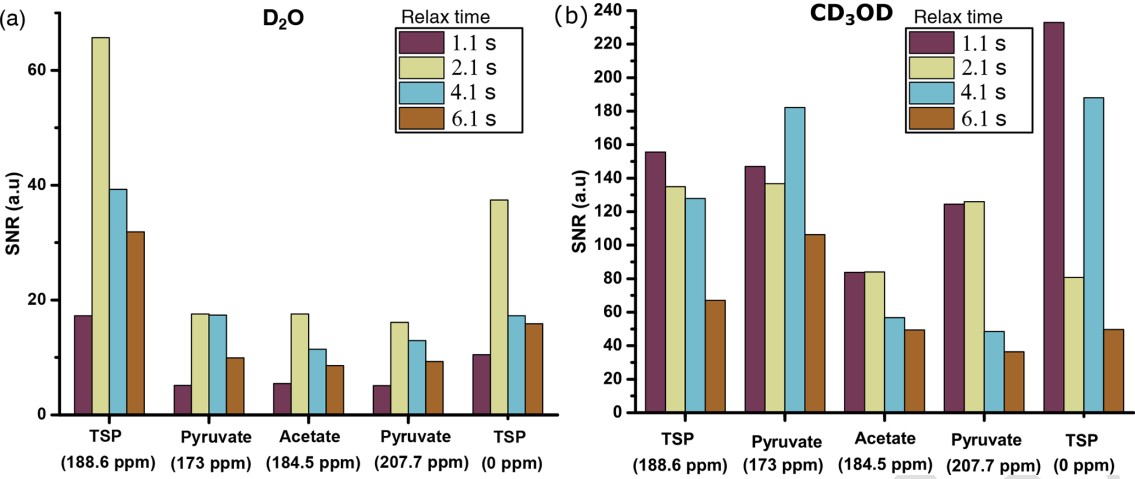

**Figure 7.** Sensitivity comparison of $^{13}$C–$\{^{1}$H$\}$ signal at different relax times using **(a)** $D_2O$ and **(b)** $CD_3OD$ as dissolution solvent. The optimum relax time with $D_2O$ and $CD_3OD$ is chosen to be 2.1 and 1.1 s.

dissolution solution. Table 2 helps to evaluate the role of longitudinal relaxation in the effect of the optimization of the dissolution time on the observed hyperpolarization in solution. Similar $T_1$ measurements in $CD_3OD$ could not be performed as metabolites at a 500 mM concentration and even at a 100 mM concentration (particularly alanine and pyruvate) were not soluble enough in $CD_3OD$. However, relative differences in $T_1$ relaxation values may be anticipated to follow similar trends as in $D_2O$. Overall, $CD_3OD$ as a dissolution solvent compared to $D_2O$ showcases superior performance by offering better line shape, which translates into improved signal sensitivity of our model metabolite mixture sample. However, for the wide range of biological samples, a lack of chemical shift database of metabolites and the inefficient solubility of metabolites in $CD_3OD$ could impose additional challenges. On the one hand, the chemical shift assignment challenge in $CD_3OD$ could be overcome by "spiking" experiments. On the other hand, previous studies showed that aqueous-solvent-based dissolution techniques can be improved using back-pressure techniques (Kouřil et al., 2021, 2019; Katsikis et al., 2015; Bowen and Hilty, 2010; Ceillier et al., 2021). Therefore, in general, the choice of the dissolution solvent between $CD_3OD$ and $D_2O$ should be weighed by considering such factors.

### 4.3.3    Transfer line optimization (c.3)

First, we suitably adjusted the length (370 cm) of sample transfer line according to the distance between the polarizer and NMR acquisition magnet by reducing the extra length of the line that was present in the initial setting. The effect of the transfer line inner diameter on the signal integral values along with the repeatability is presented in Fig. D1, which shows superior signal obtained with the small diameter ID (1.575 mm) transfer line compared to the wider one

(2.375 mm). A possible explanation for such a difference in the signal integral would be better homogeneity and smaller segregation of the liquid and gas mixture in the smaller ID of sample transfer line compared to the wider ID, resulting in a faster sample mass transfer. To maintain signal line shape repeatability, care should be taken at the connection point of the sample transfer line to the dissolution stick and injector.

## 5    Result of optimization

Finally, we have compared the metabolite signal integrals and sensitivity to investigate the performance of the two dissolution solvents with the optimized d-DNP setting and benchmarked the improvement of signal with respect to the spectrum acquired before optimization of d-DNP settings (see spectra in Fig. 8). Figure 9 showcases significant improvement in sensitivity (about 5 times improvement on quaternary $^{13}$C and 50 times improvement on protonated $^{13}$C) as well as improvement in the signal integral, especially with $CD_3OD$ compared to signals obtained using the initial parameters before DNP optimization. The main contributing factors of this improvement are the shorter dissolution duration and faster stabilization of the dissolved liquid inside the NMR tube. These factors also contributed to improving the line shape and the linewidth of $^{13}$C signals with $CD_3OD$ (see spectra in Fig. 8) significantly (at least 3 times sharper). We found that after optimization the improved sensitivity with $CD_3OD$ enables the detection and analysis of the quaternary alanine signal which was not detected before. Moreover, the overall optimization improved the limit of detection, which enabled the observation of the protonated $^{13}$C signals of metabolites at natural abundance (e.g., signals of acetate and pyruvate at 29 and 26 ppm).

We have summarized the impact of optimization in Table 3, which showcases the changes in spectral qualities of

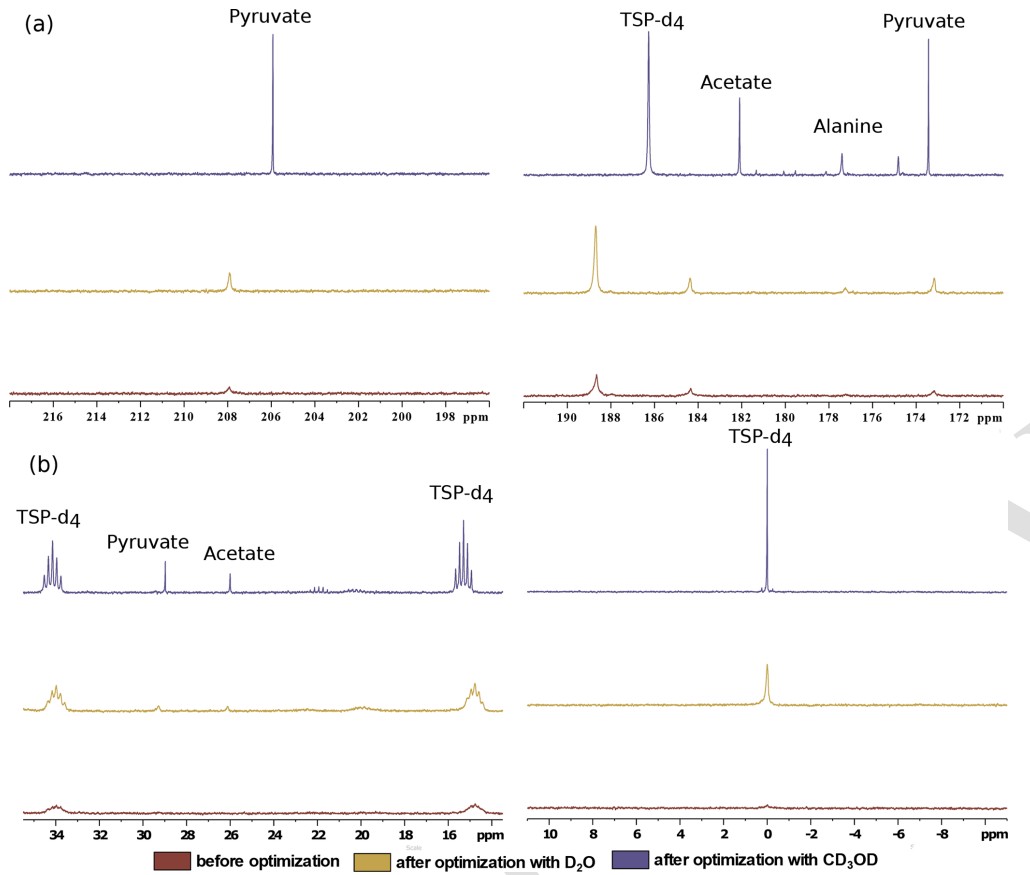

**Figure 8.** Comparison of $^{13}$C–{$^1$H} spectra of metabolites before and after optimization in the **(a)** quaternary $^{13}$C region and **(b)** protonated $^{13}$C region using two dissolution solvents.

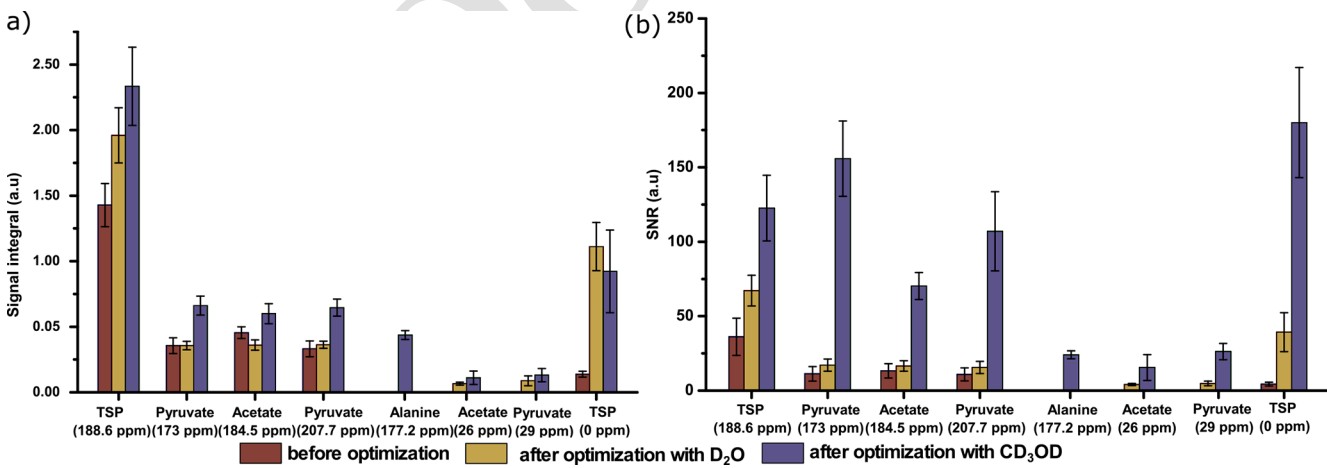

**Figure 9.** Comparison of $^{13}$C–{$^1$H} signals of metabolites with respect to **(a)** the average signal integral and **(b)** the average sensitivity along with the standard deviation, with and without systematically optimized parameters using two dissolution solvents.

signals (signal integral repeatability, linewidth, and liquid-state polarization) before and after optimization of the d-DNP setting. Table 3 highlights the significant improvement of the linewidth and liquid-state polarization (especially with CD$_3$OD).

In order to evaluate the impact of optimization on repeatability, Table 3 also compares the repeatability of absolute and normalized signal integrals (with respect to TSP signal at 188 ppm). The results demonstrate a considerable improvement for the quaternary $^{13}$C signals in both solvents after op-

**Table 3.** Repeatability comparison of $^{13}$C–{$^1$H} signal integrals of metabolites with and without systematically optimized parameters and with two different dissolution solvents.

| Metabolites | Experimental condition | Repeatability (cv %) | | Linewidth (Hz) | Liquid state polarization[c] (%) |
|---|---|---|---|---|---|
| | | Absolute | Normalized | | |
| Pyruvate (207.7 ppm) | Before | 18.2 | 6.6 | 11.6 | 8.0 |
| | After (D$_2$O) | 7.4 | 6.5 | 8.3 | 8.0 |
| | After (CD$_3$OD) | 10.1 | 3.0 | 1.8 | 15.0 |
| TSP-d$_4$ (188.6 ppm) | Before | 11.5 | _$^a$ | 13.0 | 9.0 |
| | After (D$_2$O) | 10.7 | _$^a$ | 9.1 | 14.0 |
| | After (CD$_3$OD) | 12.8 | _$^a$ | 6.6 | 17.0 |
| Acetate (184.4 ppm) | Before | 9.8 | 3.0 | 12.1 | 12.0 |
| | After (D$_2$O) | 10.9 | 3.9 | 8.7 | 11.0 |
| | After (CD$_3$OD) | 12.6 | 0.9 | 2.7 | 18.0 |
| Alanine (177.2 ppm) | Before | _$^b$ | _$^b$ | _$^b$ | _$^b$ |
| | After (D$_2$O) | _$^b$ | _$^b$ | 14.1 | 5.0 |
| | After (CD$_3$OD) | 8.0 | 5.5 | 6.7 | 9.0 |
| Pyruvate (172.6 ppm) | Before | 17.0 | 8.1 | 11.6 | 9.0 |
| | After (D$_2$O) | 9.0 | 6.2 | 8.4 | 11.0 |
| | After (CD$_3$OD) | 11.0 | 2.1 | 1.1 | 19.0 |
| Acetate (26 ppm) | Before | _$^b$ | _$^b$ | _$^b$ | _$^b$ |
| | After (D$_2$O) | 17.0 | 26.3 | 10.1 | 2.0 |
| | After (CD$_3$OD) | 46.0 | 51.0 | 2.7 | 4.0 |
| Pyruvate (28.9 ppm) | Before | _$^b$ | _$^b$ | _$^b$ | _$^b$ |
| | After (D$_2$O) | 43.0 | 50.0 | 8.2 | 3.0 |
| | After (CD$_3$OD) | 39.0 | 33.0 | 1.6 | 6.0 |
| TSP-d$_4$ (0 ppm) | Before | 16.1 | 17.3 | 13.5 | 0.4 |
| | After (D$_2$O) | 16.5 | 8.0 | 8.3 | 2.0 |
| | After (CD$_3$OD) | 34.1 | 29.7 | 1.1 | 3.0 |

$^a$ Normalized signal repeatability was obtained from the normalized signal integral with respect to TSP-d$_4$ (188.6 ppm). $^b$ Peak areas were not measured when the signals were below the limit of quantification (SNR < 10). $^c$ Liquid-state polarization was calculated by comparing the thermal signal integral at a 500 mM concentration of metabolites acquired with similar acquisition parameters as d-DNP, and the polarization values are rounded off suitably.

timization compared to the signal obtained before optimization. However, reduction of the dissolution time and stabilization delay introduces additional challenges in the manual dissolution efficiency to maintain the repeatability of the protonated $^{13}$C as the $T_1$ relaxation value of the fast-relaxing protonated $^{13}$C spins in the presence of 2 mM TEMPOL (final concentration of radical after dissolution) is about 6 s. We found that the TSP signal at 0 ppm and the protonated $^{13}$C signals of metabolites showed much higher variability. The $T_1$ value (Table 2) of protonated TSP is the minimum among all the peaks of our interest, which can be linked to the much higher associated signal variability. In future DNP-enhanced metabolomics studies, care should be taken when choosing the reference signals. $T_1$ measurements under DNP conditions could provide a hint towards the choice of a reference, as presented in Table 2. The higher variability of the protonated $^{13}$C signals of the metabolites may be linked to

their lower sensitivity, which occurs due to the combined effect of shorter relaxation times compared to TSP-d$_4$. Future optimization studies will focus on further improving the repeatability of protonated $^{13}$C signal by better controlling the repeatability of the dissolution step.

## 6 Conclusion

We have presented the detailed report of a fine, user-oriented optimization of a semi-automated, prototype d-DNP experimental setting dedicated to $^{13}$C NMR of metabolite mixtures at natural abundance. The optimization allows the scope of natural-abundance $^{13}$C metabolomics studies to be extended with high repeatability. The optimized conditions make it possible to identify the previously inaccessible protonated $^{13}$C signals of metabolites with improved line shape. Still, it also opens the way to further optimization. In the near fu-

ture, with the present d-DNP setting, it would be interesting to investigate the impact of a few parameters that would require minor modifications of the instrumental setting, such as the effect of the magnetic tunnel, the dissolution solvent volume, and the length and geometry of the injector. Further improvement of the signal repeatability of [13]C signals (especially the protonated [13]C spins) will probably require more extensive instrumental developments, such as an automated dissolution system and rapid sample transfer module. Further reduction of the dissolution, transfer, and stabilization delays could even enable the acquisition of DNP-enhanced [1]H spectra of the metabolites. Also, recent reports on the use of porous polarizing matrices could provide a tremendous boost for metabolomic applications as it makes DNP highly independent on the sample, and it removes paramagnetic relaxation in the liquid state (Cavaillès et al., 2018; El Daraï et al., 2021). Overall, we have established a series of optimization guidelines which could be of general interest for analytical applications of d-DNP NMR. We hope that such optimized d-DNP NMR setting will pave the way to new applications of hyperpolarized [13]C NMR of complex mixtures at natural abundance.

## Appendix A: $\mu w$ frequency and power optimization

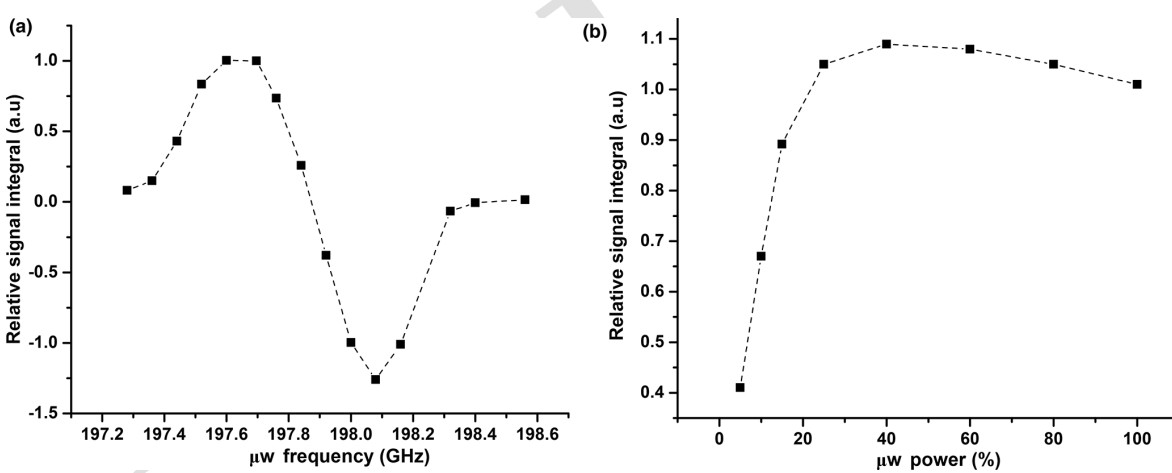

**Figure A1. (a)** Plot of relative [1]H DNP signal integrals vs. $\mu w$ frequency at 50 mM TEMPOL. Signals are normalized with respect to the signal at 197.69 GHz. **(b)** Plot of relative [1]H DNP signal integrals vs. $\mu w$ power (%). Signals are normalized with respect to the signal at 198.08 GHz with a triangular frequency modulation with a bandwidth ($\Delta f_{\mu w}$) of $\pm 5$ MHz and frequency of 10 kHz.

## Appendix B: Pulse program to monitor $^1$H DNP buildups and solid-state DNP pulse sequence via cross-polarization (CP)

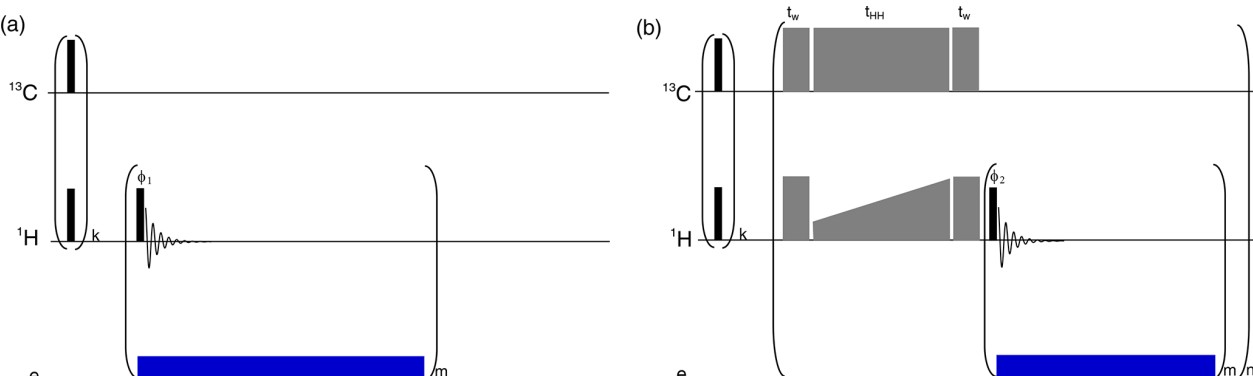

**Figure B1.** Pulse sequence used to monitor **(a)** $^1$H DNP buildups and **(b)** solid-state DNP pulse sequence via cross-polarization (CP) for d-DNP. A train ($k = 64$) of $\pi/2$ pulses was applied to both RF channels for pre-saturation. For $^1$H buildup measurements, a small-flip-angle radio frequency (RF) pulse is employed after each $\mu w$ irradiation of 10 s (depicted as "blue" block) to monitor the polarization level. Here CP (depicted as "grey" blocks) is performed using adiabatic half-passage pulses ($t_w = 175\,\mu s$) to convert longitudinal magnetization into transverse magnetization before the start of the contact pulse ($t_{HH} = 15$ ms) and vice versa after the contact. Contact pulses use RF powers of 15 W on $^1$H (using rectangular pulse with constant RF amplitudes of 21 kHz) and 60 W on $^{13}$C (using ramped up pulse with linearly increasing RF amplitudes from 16 to 23.2 kHz). In total, 16 CP contacts ($n = 16$) are made. For each contact, a sequence of $m = 4$ pulses with low-flip-angle pulse (5°) is applied on the $^{13}$C channel to monitor the buildup of the polarization from $^1$H to $^{13}$C. Microwave irradiation is selectively switched on after each CP contact for 80 s to improve DNP polarization efficiency by avoiding the significant contribution of electron spin in the nuclear spin relaxation rate. `TS5 TS6`

## Appendix C: $^1$H DNP polarization buildup plot at different radical concentrations

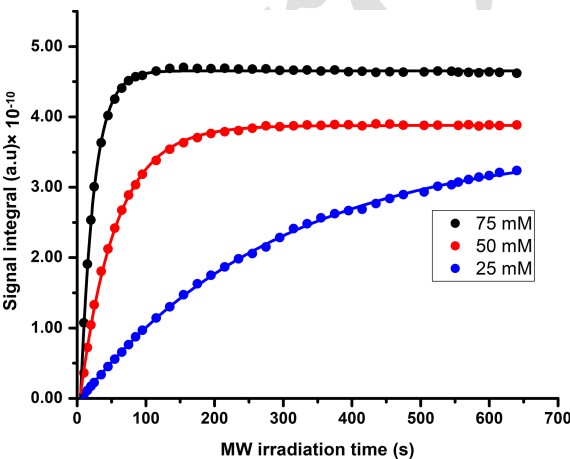

**Figure C1.** $^1$H DNP polarization buildup plot at different TEM-POL concentrations (75, 50, and 25 mM). The data points are fitted monoexponentially to obtain the DNP buildup rate.

## Appendix D: Sample transfer line optimization

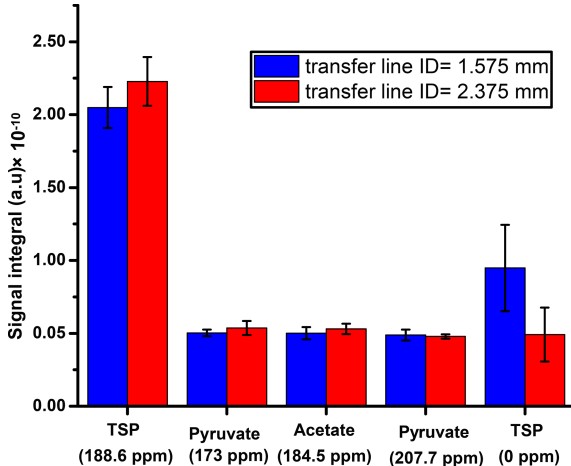

**Figure D1.** Comparative plot of liquid-state $^{13}$C signal integrals between two different sample transfer line inner diameters at the same dissolution delay.

## Appendix E: Injector and NMR tube connection

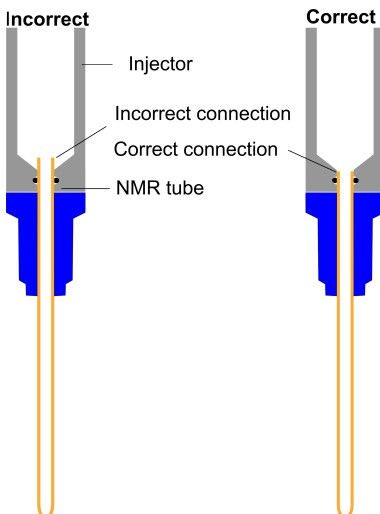

**Figure E1.** Schematic presentation of correct and incorrect connections between the top part of the NMR tube and injector.

**Data availability.** The NMR data shown in Figs. 2 to 9 are available for download in TopSpin format from https://doi.org/10.5281/zenodo.6810794 (Dey et al., 2022).

**Author contributions.** AD, BC, and VR designed experiments with the suggestions of all co-authors. AD, BC, and KL performed all the experiments. AD, BC, KL, and VR did the formal analysis of the data which were then critically discussed and validated by all the co-authors. AD did data curation and wrote the original draft. PG and JND conceptualized the research goal, analyzed the data, and edited the manuscript. All co-authors edited the manuscript and approved the final version.

**Competing interests.** Dmitry Eshchenko, Marc Schnell, Roberto Melzi, and James G. Kempf are employees of Bruker Biospin, which supplied the d-DNP polarizer. It is not a commercial instrument but a step in ongoing Bruker R&D.

**Acknowledgements.** The authors acknowledge the French National Infrastructure for Metabolomics and Fluxomics MetaboHUB-ANR-11-INBS-0010 (http://www.metabohub.fr, last access: 25 September 2022) and the Corsaire metabolomics core facility (Biogenouest). This work includes NMR experiments carried out on the CEISAM NMR platform.

**Financial support.** This research has been supported by the European Research Council, H2020 European Research Council (grant nos. SUMMIT (814747), DINAMIX (801774), HP4all (714519)), the Marie Skłodowska-Curie grant TS7 (ZULF (grant no. 766402)), the Agence Nationale de la Recherche (grant no. MetaboHUB-ANR-11-INBS-0010), the Conseil Régional des Pays de la Loire (Connect Talent/HPNMR), the French CNRS, Nantes Université, and Lyon 1 University.

**Review statement.** This paper was edited by Geoffrey Bodenhausen and reviewed by Benno Meier and one anonymous referee.

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

### Remarks from the language copy-editor

### Remarks from the typesetter

**TS2** Figure 1 has been adjusted; please check. Please note that vector graphics cannot be included in the PDFLaTeX since certain fonts or other content cannot be embedded and such content would then not show up in some browsers or *.pdf viewers. As a result, affected figures might appear incomplete to some readers. Therefore, we only include *.png and *.jpg figures in the article *.pdf. However, since we also publish all articles in full-text HTML, we will provide your vector graphics as high-resolution figures so that readers are able to download and enlarge the figures for re-use Please see https://www.hydrol-earth-syst-sci.net/23/1163/2019/ as an example. The high-resolution files can be downloaded via the "Download" button below the figure.

**TS6** Please note that the figure cannot just be replaced after the paper has been accepted for publication. According to our standards, changes like this first need to be *officially* approved by the editor in a process called post-review adjustment; this is necessary to retain transparency for the reader. Therefore, please provide the new version of Fig. B1 and a detailed explanation for the required changes. I will take care of the upload. Upon approval, the figure can be replaced. Thank you for your understanding.