# Peer review of "Fine optimization of a dissolution-DNP experimental setting for 13C NMR of metabolic samples"

_Magnetic Resonance, 2022_

## Author Comment (AC1)

**Response to referees for the paper "Fine optimization of a dissolution-DNP experimental setting for $^{13}$C NMR of metabolic samples" – A. Dey *et al.***

We are grateful to both referees for their insightful comments and suggestions. A revised manuscript will be prepared based on their remarks, as described in details below.

**Referee 1:**

I will below suggest the most pressing things to work with and at the same time declare that I at this stage did not go into details.

**High attention:**

1. In all figures the "alanine" chemical shift is given as 173 ppm in D2O. The pH is not stated but judged relative to the chemical shifts for the other metabolites this is not correct. It should be closer to 177 ppm. Pyruvate (n.a.) will show two chemical shifts in the carbonyl region and the 173 ppm fits well with C1 for pyruvate. Did the authors assign the spectra correct? Looking at the spectra given in figure 8.a then the red spectrum "optimized with D2O" shows pyruvate C2 at 208 ppm, TSP-d4 at 189 ppm, acetate at 184 ppm, alanine (I suggest) at 177 ppm and pyruvate C1 at 173 ppm. In methanol (green spectrum) there are one additional signal and some very small signals (could be relevant). Did the authors measure the metabolites individually with thermal NMR in methanol to verify chemical shifts?

Answer: Our initial assignments were made based on the literature. However, we have rechecked the assignment of each metabolites in D$_2$O by acquiring the spectra at higher concentration of metabolites under thermal conditions with sequential addition of the metabolites and keeping the similar solvent composition as in the resultant DNP solution. Also, we have performed spiking experiments with dissolution DNP: one spectrum was recorded with all metabolites at a 5 mM concentration and a second one where the concentration of pyruvate was increased and keeping rest of the other metabolites at 5 mM. Both experiments indicate that the initial assignment of quaternary alanine signal was wrong. It should be at 177.26 ppm. And the peak at 173.2 ppm belongs to Pyruvate. We will modify this in the manuscript accordingly. We thank the referee for spotting this mistake.

2. Most figures have an odd y-axis unit. I suggest to leaving out "x10-10" with the a.u being sufficient, and to standardize the axis units (ex. fig. 4b). It is important that what should be comparable can be compared. Ex. Should figure 3a not be comparable to fig.4a? e.g. the red stables in fig.4a for 1H DNP (uw50%) are not comparable to the red (50mM) 1H DNP (uw 50%)?

Answer: We will modify the y-axis unit in the revised manuscript. There is a mistake in the figure 3a scaling in the y-axis value (it should be $\times 10^{-11}$ instead of $\times 10^{-10}$) which will be corrected as well.

Also, in the solid-state figures I do not understand how to interpret the "thermal signal" (in fig.3a I guess it is as in fig4.a the 1H signal without microwaves on. But how can this signal be on the same scale as the 1H DNP signal? In that case it looks as though the polarization is very low.

Answer: In our case the $^1$H "thermal signal" is acquired to compare the variation of amount of the sample taken in the sample cup among different identical samples, not to acquire actual signal under thermal conditions. Here we acquired signals without subtracting the signal from the empty sample cup (background signal). Before acquiring the DNP signal, we have saturated the signals from the

background and the actual thermal signal as indicated in the pulse sequence (figureB1). We will mention this in the manuscript and remove the term "thermal signal" to avoid ambiguity.

Make sure to give all the important information in the figure legends eg. tempol concentration in figure 4.

Answer: The TEMPOL concentration is 50 mM, Figure 4 legend will be corrected.

3. Figure 6 should include protonated carbons (shown in fig.8). As should Table 2.

Answer: In the conditions of Figure 6 as well as for table 2, the sensitivity of protonated carbon signals obtained with $D_2O$ as a dissolution solvent are below the limit of quantification (SNR<10). Therefore, it is not feasible to determine the integral value of such spectra. This will be explained in the revised manuscript.

4. Why is the longer "relax" times for "TSP 0 ppm" (blue and green) in figure 5b not comparable to similar signal quantifications in figure 2b? Is it not TSPd4 in all experiments? (should be stated).

Answer: There is no Figure 2b. We are not sure which figure to compare. Indeed, we used TSP-$d_4$ (mentioned in line 135) for all of our experiments.

5. The high variability of the "TSP 0 ppm" signal in methanol (fig. 7b and Table 2) should be discussed in relation to the use of this standard for relative quantification or absolute quantification and as chemical shift reference.

Answer: We will add a line in the revised manuscript to explain that inn future DNP enhanced metabolomics study, care should be taken when choosing the reference molecules. $T_1$ measurements under DNP conditions could provide a hint towards the choice of reference, as discussed below. In our case $T_1$ value of protonated TSP is the minimum among all the peaks of our interest which contributes to the high variability.

6. To be able to discuss the impact of the different optimized parameters it would be valuable to measure the $T_1$ of the different carbons in the included metabolites and for the TSP standard. This can be done straightforwardly by increasing the concentration of the metabolites to 50 or 100 mM in a simulated sample (50 mM PA in 6:3:1 glycerol:D2O:H2O, total vol. 200 ul dissolved in 5 ml methanol) and run 2 inversion recovery experiments -one for carbonyl carbons and one for aliphatic carbons. It would be interesting to also perform these experiments without the added tempol radical.

Answer: As suggested by the reviewer, we have measured the $T_1$ value in the presence and absence of TEMPOL as presented in the following Table (that will be provided in the revised version). These $T_1$ measurement were done by dissolving the metabolites (each at 500 mM concentration) in $D_2O$ as suggested by the reviewer. The $T_1$ measurement in $CD_3OD$ is not feasible as metabolites at a 500 mM concentration and even at a 100 mM concentration (particularly alanine and pyruvate) are not soluble enough in $CD_3OD$. However, relative differences in $T_1$ relaxation value may be anticipated to follow similar trends as in $D_2O$. In particular, this table shows the much lower $T_1$ of the TSP peak at 0 ppm, which can be linked to the much higher associated signal variability, as discussed above.

| Metabolites and chemical shift | | Pyruvate (207.7 ppm) | TSP-d$_4$ (188.6 ppm) | Acetate (184.4 ppm) | Alanine (177.2 ppm) | Pyruvate (172.6 ppm) | Acetate (26 ppm) | Pyruvate (28.9 ppm) | TSP-d4 (0 ppm) |
|---|---|---|---|---|---|---|---|---|---|
| $T_1$ (s) | With TEMPOL | 13.6 | 6.1 | 12.4 | 12.3 | 14.3 | 6.2 | 6.4 | 3 |
| | Without TEMPOL | 22.8 | 35.2 | 51.8 | 28.2 | 41 | 10.4 | 11.7 | 5.2 |

7. Since this is a hyperpolarization method optimization paper it is relevant to measure the polarization in a liquid state sample. To save time this is most easily done using a condition matched external standard with an exact concentration, ex. use 1-13C-acetate which can be made reliably in high concentration. The measurement will not be decimal exact but this is not important.

Answer: As suggested, we measured the polarization of the liquid state by comparing the thermal signal at a 500 mM concentration of metabolites. We will introduce the polarization value in Table 2 of the revised manuscript. The polarization values of the metabolites are as follows:

| Metabolites and chemical shift | Pyruvate (207.7 ppm) | Acetate (184.4 ppm) | Alanine (177.2 ppm) | Pyruvate (172.6 ppm) | Acetate (26 ppm) | Pyruvate (28.9 ppm) |
|---|---|---|---|---|---|---|
| Polarization (%) | 14.6 | 17.6 | 9.4 | 18.6 | 3.6 | 5.8 |

Other points:

**p.8 section on "B.4 Vitrification parameters":**

It is natural when working with complicated methods that experimental routines are implemented that has little theoretical meaning. Several points in this section refer to such experimental routines based on non-investigated observations. If rate of vitrification is important it should be shown. If it matters in which order the metabolites are dissolved (water first or water:glycerol mix or glycerol) it is a matter of solubility and should be investigated. Then also sample temperature may be an issue as well as total dissolution volume. I suggest you separate out parameters that you have identified as possibly important for later study/optimization from the parameters that you have investigated and can conclude on and discuss.

Answer: We have discussed the order of sample preparation in section A.2, and section B.2 already mentions that the rate of vitrification did not impact our result. We will try to make these points clearer in the revised manuscript. As suggested by the reviewer, we will add a few lines describing the parameters that could be important to investigate in further studies.

**Discussion:**
The discussion is generally kept to stating the findings with a comment. The results are rarely discussed. Ex.: The authors have previously published (also nicely referenced in the manuscript)

significant contributions to the use of dDNP NMR for allowing 13C direct detect natural abundance mixture analysis. Significant findings in those reports are not discussed relative to the results presented in this manuscript (e.g. use of Hellmanex and a suited internal standard for quantification). Please discuss the alternative choices in this manuscript and how they have improved previous results or was not part of the purpose.

The results are summarized stating that the main contribution to the significant method improvement is the transfer time. It would be interesting with a discussion about the consequences of the improvements. Especially the important choice of dissolving in methanol could be strengthen with a discussion on chemical shift changes in methanol (lack of database, temperature and concentration influences).

Answer: In the revised manuscript, we will extend the discussion based on the reviewer's suggestions.

**I just noticed:**

Spelling error in Figure 1: 'magentic' should be 'magnetic'

Answer: It will be corrected in the revised manuscript.

Example of unprecise language: l.148 'to trace the amount' - maybe to weigh?

Answer: It will be corrected in the revised manuscript

Please explain how the factor 2900 difference in sensitivity between 13C and 1H is calculated

Answer: $\left(\frac{\gamma_H}{\gamma_C}\right)^{3-1/2}\frac{1}{0.011} = \left(\frac{\gamma_H}{\gamma_C}\right)^{5/2}\frac{1}{0.011} = 2900$ ; Progress in Nuclear Magnetic Resonance Spectroscopy, vol. 12, no. 1, pp. 41–77, 1978

**Referee 2:**

The study is empirical and one cannot argue with the findings. I do however have a few comments regarding the presentation:

- The authors have chosen to first discuss all the paramters they optimize (section 3 Experiments and parameters), and then give the results of each optimization in another section (section 4 Results and Discussion).

This structure leads to a rather long manuscript, and so the results of the optimization should be summarized in a table, stating resolution before and after the optimisation, as well as the polarisation levels that were obtained in the liquid state, before and after optimisation. The final values of course should be stated for each of the two solvents D2O and Methanol-d4.

Answer: As suggested, we will include linewidths, polarization in the liquid state, SNR in Table 2 of the revised manuscript.

- It is well known that high-resolution spectra can be recorded using aqueous solvents if a back-pressure technique is used (see, e.g., the works by Bowen / Hilty, and Katsikis / Günther). These works should be discussed. (BTW our group has implemented a similar back-pressure technique for the bullet system.)

Answer: While a detailed discussion of several dissolution methods is beyond the scope of this study, a few lines will be added in the revised manuscript.

- Figure 1 suggests that the field during the sample transfer never drops below 0.56 T. This is probably not correct.

Answer: We will modify Figure 1 accordingly.

- In section C1 the authors write that the heat transfer coefficient is different for different solvents. Can they give numbers? Perhaps naively I would have thought that the heat capacity of the solvents plays a more important role.

Answer: By heat transfer coefficient we meant Specific heat capacity $\times \Delta T$. The specific heat capacity of Water and methanol is about 4.18 kJ/kg K and 2.53 kJ/kg K respectively. For $CD_3OD$ and $D_2O$ we set the temperature in the dissolution oven is 156 °C and 170 °C respectively. The values will be included into the manuscript.

- Figure 3 plots the thermal signal, and the 1H DNP signal, as well as the 13C DNP signal in arbitrary units. I assume that the "thermal signal" is the 1H thermal signal (if so, this should be stated explicitly).

In that case, the 1H DNP signal is only 15 times larger than the thermal signal. What is the estimate for the proton background?

Answer: Qualitatively, the $^1$H back-ground signal integral is in the similar range of signal integral presented as "thermal signal" in the manuscript. As discussed in response to the first reviewer, we will remove the term "thermal signal".

- Negative DNP shows larger polarization (Fig. A1 a), but the authors opted for positive DNP. Why? Was the dependence of DNP enhancement on sweep width studied previously? If so, this study should be cited explicitly.

Answer: Indeed, we have opted for negative DNP as mentioned in line no 388. We will explain this in the manuscript. About the MW modulation, the corresponding study is already cited at line no 194.

- The results in table 2 should be rounded appropriately.

Answer: This will be done in the revised manuscript

My other comments are minor language corrections.

Answer: All the suggested changes will be incorporated into the revised manuscript.

- p1 line 13: relies on 1D instead of rely (analysis is singular).

- line 29: unparalleled instead of unparallel?

- p2 line 42: of 13C signal /detection/

- line 51: the references apply to dissolution-DNP, so the line should read ... such as Dissolution Dynamic ...

- line 56: "dissolution state DNP" should read dissolution-DNP

- p3 line 83: provides instead of provided

- line 97f: this sentence should be split into two

- line 101: during /the/ d-DNP experiment

- p4: Magnetic instead of Magentic

- p5. 112: identify instead of identified

- p6. l 153: /The/ PA plays /a/ central part..  A broad variety of PAs is available

- p7. l 164: the efficiency ... depend/s/

- p8. l. 211: B1a 1.2 K should probably read B1a at 1.2 K.

- p9. l. 236: in detail instead of in details.

- l. 240: pressurized instead of pressured

- p10. l263: the ... delays... contain (no s)

- l. 267: we focused on instead of in.

- l. 271: all parameters listed instead of each parameter enlisted

- l. 286: "smaller" is better than "less efficient"

- l. 295: mL with captial L, missing period at end of line

- p12 l. 332: How much is 11 % in Hertz?

- p16 line 435: the words "13C signal" appear twice.

- I don't understand the sentence in line 436ff. Is this a sensitivity vs resolution tradeoff?

- l. 439: what about the irregularity in TSP?

- p17 l. 449: this sentence should read "This often results in failure of signal acquisition.

---

## Author Response (AR1)

**Response to referees for the paper "Fine optimization of a dissolution-DNP experimental setting for $^{13}$C NMR of metabolic samples" – A. Dey** *et al.*

We are grateful to both referees for their insightful comments and suggestions. A revised manuscript will be prepared based on their remarks, as described in details below.

**Referee 1:**

I will below suggest the most pressing things to work with and at the same time declare that I at this stage did not go into details.

**High attention:**

1. In all figures the "alanine" chemical shift is given as 173 ppm in D2O. The pH is not stated but judged relative to the chemical shifts for the other metabolites this is not correct. It should be closer to 177 ppm. Pyruvate (n.a.) will show two chemical shifts in the carbonyl region and the 173 ppm fits well with C1 for pyruvate. Did the authors assign the spectra correct? Looking at the spectra given in figure 8.a then the red spectrum "optimized with D2O" shows pyruvate C2 at 208 ppm, TSP-d4 at 189 ppm, acetate at 184 ppm, alanine (I suggest) at 177 ppm and pyruvate C1 at 173 ppm. In methanol (green spectrum) there are one additional signal and some very small signals (could be relevant). Did the authors measure the metabolites individually with thermal NMR in methanol to verify chemical shifts?

Answer: Our initial assignments were made based on the literature. However, we have rechecked the assignment of each metabolites in D$_2$O by acquiring the spectra at higher concentration of metabolites under thermal conditions with sequential addition of the metabolites and keeping the similar solvent composition as in the resultant DNP solution. Also, we have performed spiking experiments with dissolution DNP: one spectrum was recorded with all metabolites at a 5 mM concentration and a second one where the concentration of pyruvate was increased and keeping rest of the other metabolites at 5 mM. Both experiments indicate that the initial assignment of quaternary alanine signal was wrong. It should be at 177.26 ppm. And the peak at 173.2 ppm belongs to Pyruvate. We have modified this in the manuscript accordingly. We thank the referee for spotting this mistake. Corrections on the figures are made on page no 12, 13, 15, 16, 17, 19.

2. Most figures have an odd y-axis unit. I suggest to leaving out "x10-10" with the a.u being sufficient, and to standardize the axis units (ex. fig. 4b). It is important that what should be comparable can be compared. Ex. Should figure 3a not be comparable to fig.4a? e.g. the red stables in fig.4a for 1H DNP (uw50%) are not comparable to the red (50mM) 1H DNP (uw 50%)?

Answer: We have modified the y-axis unit in the revised manuscript. There is a mistake in the figure 3a scaling in the y-axis value (it should be $\times 10^{-11}$ instead of $\times 10^{-10}$) which is corrected as well. Corrections on the figures are made on page no 13, 14, 15, 16, 19.

Also, in the solid-state figures I do not understand how to interpret the "thermal signal" (in fig.3a I guess it is as in fig4.a the 1H signal without microwaves on. But how can this signal be on the same scale as the 1H DNP signal? In that case it looks as though the polarization is very low.

Answer: In our case the $^1$H "thermal signal" is acquired to compare the variation of amount of the sample taken in the sample cup among different identical samples, not to acquire actual signal under thermal conditions. Here we acquired signals without subtracting the signal from the empty sample cup (background signal). Before acquiring the DNP signal, we have saturated the signals from the

background and the actual thermal signal as indicated in the pulse sequence (figureB1). We have mentioned this in the manuscript and remove the term "thermal signal" to avoid ambiguity. Text added at page no 6.

Make sure to give all the important information in the figure legends eg. tempol concentration in figure 4.

Answer: The TEMPOL concentration is 50 mM, figure 4 legend at page no 14 is corrected.

3. Figure 6 should include protonated carbons (shown in fig.8). As should Table 2.

Answer: In the conditions of Figure 6 as well as for table 2, the sensitivity of protonated carbon signals obtained with $D_2O$ as a dissolution solvent are below the limit of quantification (SNR<10). Therefore, it is not feasible to determine the integral value of such spectra. This is explained in the revised manuscript. Explanation added at present table no 3 in page 20.

4. Why is the longer "relax" times for "TSP 0 ppm" (blue and green) in figure 5b not comparable to similar signal quantifications in figure 2b? Is it not TSPd4 in all experiments? (should be stated).

Answer: There is no Figure 2b. We are not sure which figure to compare. Indeed, we used TSP-$d_4$ (mentioned in line 135) for all of our experiments.

5. The high variability of the "TSP 0 ppm" signal in methanol (fig. 7b and Table 2) should be discussed in relation to the use of this standard for relative quantification or absolute quantification and as chemical shift reference.

Answer: We have added a line in the revised manuscript to explain that inn future DNP enhanced metabolomics study, care should be taken when choosing the reference molecules. $T_1$ measurements under DNP conditions could provide a hint towards the choice of reference, as discussed below. In our case $T_1$ value of protonated TSP is the minimum among all the peaks of our interest which contributes to the high variability. Text added at page no 21.

6. To be able to discuss the impact of the different optimized parameters it would be valuable to measure the $T_1$ of the different carbons in the included metabolites and for the TSP standard. This can be done straightforwardly by increasing the concentration of the metabolites to 50 or 100 mM in a simulated sample (50 mM PA in 6:3:1 glycerol:D2O:H2O, total vol. 200 ul dissolved in 5 ml methanol) and run 2 inversion recovery experiments -one for carbonyl carbons and one for aliphatic carbons. It would be interesting to also perform these experiments without the added tempol radical.

Answer: As suggested by the reviewer, we have measured the $T_1$ value in the presence and absence of TEMPOL as presented in the following Table (provided in the revised version). These $T_1$ measurement were done by dissolving the metabolites (each at 500 mM concentration) in $D_2O$ as suggested by the reviewer. The $T_1$ measurement in $CD_3OD$ is not feasible as metabolites at a 500 mM concentration and even at a 100 mM concentration (particularly alanine and pyruvate) are not soluble enough in $CD_3OD$. However, relative differences in $T_1$ relaxation value may be anticipated to follow similar trends as in $D_2O$. In particular, this table shows the much lower $T_1$ of the TSP peak at 0 ppm, which can be linked to the much higher associated signal variability, as discussed above. Table (new table no 2) added at page no 17.

| Metabolites and chemical shift | | Pyruvate (207.7 ppm) | TSP-d4 (188.6 ppm) | Acetate (184.4 ppm) | Alanine (177.2 ppm) | Pyruvate (172.6 ppm) | Acetate (26 ppm) | Pyruvate (28.9 ppm) | TSP-d4 (0 ppm) |
|---|---|---|---|---|---|---|---|---|---|
| $T_1$ (s) | With TEMPOL | 13.6 | 6.1 | 12.4 | 12.3 | 14.3 | 6.2 | 6.4 | 3 |
| | Without TEMPOL | 22.8 | 35.2 | 51.8 | 28.2 | 41 | 10.4 | 11.7 | 5.2 |

7. Since this is a hyperpolarization method optimization paper it is relevant to measure the polarization in a liquid state sample. To save time this is most easily done using a condition matched external standard with an exact concentration, ex. use 1-13C-acetate which can be made reliably in high concentration. The measurement will not be decimal exact but this is not important.

Answer: As suggested, we measured the polarization of the liquid state by comparing the thermal signal at a 500 mM concentration of metabolites. We have introduced the polarization value in Table 2 of the revised manuscript. The polarization values of the metabolites are as follows:

| Metabolites and chemical shift | Pyruvate (207.7 ppm) | Acetate (184.4 ppm) | Alanine (177.2 ppm) | Pyruvate (172.6 ppm) | Acetate (26 ppm) | Pyruvate (28.9 ppm) |
|---|---|---|---|---|---|---|
| Polarization (%) | 14.6 | 17.6 | 9.4 | 18.6 | 3.6 | 5.8 |

Table 3 is added at page no 20.

Other points:

**p.8 section on "B.4 Vitrification parameters":**

It is natural when working with complicated methods that experimental routines are implemented that has little theoretical meaning. Several points in this section refer to such experimental routines based on non-investigated observations. If rate of vitrification is important it should be shown. If it matters in which order the metabolites are dissolved (water first or water:glycerol mix or glycerol) it is a matter of solubility and should be investigated. Then also sample temperature may be an issue as well as total dissolution volume. I suggest you separate out parameters that you have identified as possibly important for later study/optimization from the parameters that you have investigated and can conclude on and discuss.

Answer: We have discussed the order of sample preparation in section A.2, and section B.2 already mentions that the rate of vitrification did not impact our result. We have tried to make these points clearer in the revised manuscript. As suggested by the reviewer, we have added a few lines describing the parameters that could be important to investigate in further studies. Text added at page no 8 and 21.

**Discussion:**

The discussion is generally kept to stating the findings with a comment. The results are rarely discussed. Ex.: The authors have previously published (also nicely referenced in the manuscript) significant contributions to the use of dDNP NMR for allowing 13C direct detect natural abundance mixture analysis. Significant findings in those reports are not discussed relative to the results presented in this manuscript (e.g. use of Hellmanex and a suited internal standard for quantification). Please discuss the alternative choices in this manuscript and how they have improved previous results or was not part of the purpose.

The results are summarized stating that the main contribution to the significant method improvement is the transfer time. It would be interesting with a discussion about the consequences of the improvements. Especially the important choice of dissolving in methanol could be strengthen with a discussion on chemical shift changes in methanol (lack of database, temperature and concentration influences).

Answer: In the revised manuscript, we have extended the discussion based on the reviewer's suggestions. Text added at page no 17 and 18.

**I just noticed:**

Spelling error in Figure 1: 'magentic' should be 'magnetic'

Answer: It is corrected in the revised manuscript. (page no 4)

Example of unprecise language: l.148 'to trace the amount' - maybe to weigh?

Answer: It is corrected in the revised manuscript. (page no 6)

Please explain how the factor 2900 difference in sensitivity between 13C and 1H is calculated

Answer: $\left(\frac{\gamma_H}{\gamma_C}\right)^{3-1/2}\frac{1}{0.011} = \left(\frac{\gamma_H}{\gamma_C}\right)^{5/2}\frac{1}{0.011} = 2900$ ; Progress in Nuclear Magnetic Resonance Spectroscopy, vol. 12, no. 1, pp. 41–77, 1978

**Referee 2:**

The study is empirical and one cannot argue with the findings. I do however have a few comments regarding the presentation:

- The authors have chosen to first discuss all the paramters they optimize (section 3 Experiments and parameters), and then give the results of each optimization in another section (section 4 Results and Discussion).

 This structure leads to a rather long manuscript, and so the results of the optimization should be summarized in a table, stating resolution before and after the optimisation, as well as the polarisation levels that were obtained in the liquid state, before and after optimisation. The final values of course should be stated for each of the two solvents D2O and Methanol-d4.

Answer: As suggested, we have included linewidths, polarization in the liquid state, SNR in Table 3 of the revised manuscript. Table 3 included at page no 20.

- It is well known that high-resolution spectra can be recorded using aqueous solvents if a back-pressure technique is used (see, e.g., the works by Bowen / Hilty, and Katsikis / Günther). These works should be discussed. (BTW our group has implemented a similar back-pressure technique for the bullet system.)

Answer: While a detailed discussion of several dissolution methods is beyond the scope of this study, a few lines are added in the revised manuscript at page no 18.

- Figure 1 suggests that the field during the sample transfer never drops below 0.56 T. This is probably not correct.

Answer: We have modified Figure 1accordingly. (page no 4)

- In section C1 the authors write that the heat transfer coefficient is different for different solvents. Can they give numbers? Perhaps naively I would have thought that the heat capacity of the solvents plays a more important role.

Answer: By heat transfer coefficient we meant Specific heat capacity $\times$ $\Delta$T. The specific heat capacity of Water and methanol is about 4.18 kJ/kg K and 2.53 kJ/kg K respectively. For $CD_3OD$ and $D_2O$ we set the temperature in the dissolution oven is 156 °C and 170 °C respectively. The values are included into the manuscript. Text added at page no 10.

- Figure 3 plots the thermal signal, and the 1H DNP signal, as well as the 13C DNP signal in arbitrary units. I assume that the "thermal signal" is the 1H thermal signal (if so, this should be stated explicitly).

In that case, the 1H DNP signal is only 15 times larger than the thermal signal. What is the estimate for the proton background?

Answer: Qualitatively, the $^1$H back-ground signal integral is in the similar range of signal integral presented as "thermal signal" in the manuscript. As discussed in response to the first reviewer, we have removed the term "thermal signal". Text added at page no 6.

- Negative DNP shows larger polarization (Fig. A1 a), but the authors opted for positive DNP. Why? Was the dependence of DNP enhancement on sweep width studied previously? If so, this study should be cited explicitly.

Answer: Indeed, we have opted for negative DNP as mentioned in line no 394. We have explained this in the manuscript. About the MW modulation, the corresponding study is already cited at line no 196. Text added at page no 14.

- The results in table 2 should be rounded appropriately.

Answer: This is done in the revised manuscript. The new table 3 at page no 20.

My other comments are minor language corrections.

Answer: All the suggested changes are incorporated into the revised manuscript.

- p1 line 13: relies on 1D instead of rely (analysis is singular). (page no 1)

- line 29: unparalleled instead of unparallel? (Page no 1)

- p2 line 42: of 13C signal /detection/ (Page no 2)

- line 51: the references apply to dissolution-DNP, so the line should read ... such as Dissolution Dynamic ... (Page no 2)

- line 56: "dissolution state DNP" should read dissolution-DNP (Page no 2)

- p3 line 83: provides instead of provided (Page no 3)

- line 97f: this sentence should be split into two (Page no 3)

- line 101: during /the/ d-DNP experiment (Page no 3)

- p4: Magnetic instead of Magentic (Page no 4)

- p5. 112: identify instead of identified (Page no 5)

- p6. l 153: /The/ PA plays /a/ central part..  A broad variety of PAs is available (Page no 6)

- p7. l 164: the efficiency ... depend/s/ (Page no 7)

- p8. l. 211: B1a 1.2 K should probably read B1a at 1.2 K. (Page no 8)

- p9. l. 236: in detail instead of in details. (Page no 9)

- l. 240: pressurized instead of pressured (Page no 9)

- p10. l263: the ... delays... contain (no s) (Page no 10)

- l. 267: we focused on instead of in. (Page no 10)

- l. 271: all parameters listed instead of each parameter enlisted (Page no 10)

- l. 286: "smaller" is better than "less efficient"(Page no 10)

- l. 295: mL with captial L, missing period at end of line (Page no 11)

- p12 l. 332: How much is 11 % in Hertz? (Page no 20)

- p16 line 435: the words "13C signal" appear twice.  (Page no 16)

- I don't understand the sentence in line 436ff. Is this a sensitivity vs resolution tradeoff?  (Page no 16)

- l. 439: what about the irregularity in TSP? (Page no 16)

- p17 l. 449: this sentence should read "This often results in failure of signal acquisition. (Page no 17)